# The Late Quaternary climate impact on the genome of the woodland strawberry (*Fragaria vesca*), a perennial herb

Tuomas Toivainen [1], J. Sakari Salonen [2], Jonathan Kirshner [3], Sergei Lembinen[1], Hanne De Kort [4], Annina Lyyski[5], Patrick P. Edger [6,7], Hrannar Smári Hilmarsson [8], Jón Hallsteinn Hallsson [9], Daniel J. Sargent[10], Klaus Olbricht[11], José F. Sánchez-Sevilla [12], Laura Jaakola[13], Johan A. Stenberg [14], Boris Duralija [15], Juozas Labokas[16,17], Henry Väre[18], Jarkko Salojärvi [19,20,21], Petri Auvinen [5], David Posé[22], Victor A. Albert [3] & Timo Hytönen [1,10] ✉

Genomes record past climatic impact on species' range shifts, admixture, refugial isolation, and adaptative evolution. However, these processes are poorly understood in perennial herbaceous species forming a dominant group of temperate flora. We present a demographic history of the perennial herb woodland strawberry (*Fragaria vesca* L.) reconstructed from 200 genomes spanning most of its European range. Temporal population structure reveals a strong division into western and eastern genetic clusters along a longitudinal climatic gradient, with eastern core populations showing greater resilience during glaciations. Divergence patterns indicate that postglacial recolonization of western and eastern Europe occurred from distinct refugia in multiple waves. The current largest, admixed populations from the Mediterranean to northern Europe form a continuous chain maintained by east–west gene flow through Central Europe, with historical migration patterns indicating comparable connections during earlier interglacials. Our reconstruction of woodland strawberry's climatic history with high temporal resolution reveals how the late Pleistocene core-periphery dynamics shaped its survival and genome evolution under climate change. The data points to populations that are essential for maintaining the long term genetic diversity of the species and opens new avenues to understand climatic adaptation of temperate flora.

The Earth underwent marked glacial-interglacial (GI) cycles during the Quaternary period, spanning the past 2.58 million years[1]. These cyclical warming/cooling climate changes caused repeated range expansions and contractions in numerous species[2–4]. Exploring these paleogeoclimatic events that have shaped the current spatial population genetic structures of species is crucial for predicting their responsiveness to future climatic events, for identifying the drivers of adaptation, and for developing biodiversity conservation strategies to ensure long-term species survival. Historically, evidence for the effects of GI-cycles on species' histories has been derived from dating of paleoecological records, such as fossil pollen preserved in sediments[3–5]. However, because plant tissue degrades relatively easily, these records are often sparse. Population genetics, a field advanced by recent technical and methodological revolutions, has become a powerful tool for investigating genetic diversity and demographic histories of species across past climatic events (hereafter, "climatic histories") in both animals[6–10] and

plants[11–16]. With the advent of whole-genome sequencing, the climatic histories of species can now be reconstructed at much higher resolution, providing novel insights into their demographic trajectories. In this study, we applied this approach to infer the historical population dynamics of woodland strawberry (*Fragaria vesca* ssp. *vesca* L.).

Perennial plants, which unlike annuals must endure extreme temperatures year-round, hold promise for recording long term population genomic signatures relevant across their extended generation times. Recent studies on woody perennials, including wild grapevine (*Vitis vinifera* ssp. *sylvestris* (Gmelin) Hegi)[14] and wild apple (*Malus sieversii* (Ledeb.) M. Roem)[12], have revealed significant declines in effective population sizes ($N_E$) during the last glacial periods, i.e., the Last Glacial Maximum (LGM, 22–17 thousand years ago (ka)), and the Penultimate Glacial Period (PGP, 190–130 ka)[1]. However, several forest tree species with high $N_E$ have shown resilience to climatic fluctuations throughout the Quaternary period[15].

Accordingly, comparing systems with sensitivity versus resilience to GI cycles can be expected to provide valuable insights into evolutionary and ecological distinctions among perennials within temperate floras.

Although perennial herbs comprise the largest fraction of plant species on Earth, with an increasing proportion of species towards colder environments[17,18], their spatial genetic structures and climatic histories remain largely unknown, as whole-genome studies encompassing broad geographic ranges are still rare. Previous studies analyzing alpine rockcress (*Arabis alpina* L.) samples from 17 sites and lyrate rockcress (*Arabidopsis lyrata* L.) from four populations, revealed strong population structures and stable climatic histories in these species. Their northern populations displayed highly reduced genetic diversity, indicating colonization-associated founder effects[19,20]. Within the genus *Fragaria* L., creamy strawberry (*Fragaria viridis* Weston), the species with the highest estimated $N_E$ to date, showed greater resilience to the LGM climatic conditions than other species[13].

In this study, we adopt the perennial herb woodland strawberry as a model and leverage whole-genome sequencing data to investigate population structure through time and its association with GI climatic history based on sampling of 200 accessions spanning most of the species' European range. This major crop wild-relative in the Rosaceae family thrives in diverse habitats, including forests, meadows, and disturbed areas such as roadsides, and has a broad geographic distribution across Eurasia and North America[21]. Woodland strawberry reproduces sexually through both outcrossing and self-fertilization, and asexually via above-ground stolons[22]. Self-fertilization and asexual reproduction are expected to increase genetic structure and inbreeding within the species. In the context of climatic history, predominantly outcrossing (and typically larger) populations are expected to be more resilient to glacial periods although asexual reproduction may confer a short-term survival advantage[23]. In woodland strawberry, sexual reproduction begins under short-day conditions in autumn with the formation of flower buds, which develop into flowers and fruits the following growing season[24,25]. Birds and mammals consume the fruits and disperse seeds[26–28], thereby promoting efficient long-distance dispersal. This animal mediated seed dispersal likely increases gene flow between populations, contributing to greater genetic connectivity across the species' range[29]. In this study, we use genomic haplotype data to reconstruct the demographic history of woodland strawberry in Europe, revealing late Pleistocene isolation-recontact dynamics across GI cycles, and Holocene colonization that have preserved distinct eastern and western genetic clusters within the species.

## Results and discussion

We assembled a collection of 200 woodland strawberry accessions spanning the species' latitudinal range in Europe (Fig. 1A)[21] and sequenced their genomes to a mean depth of coverage of 17.3. We identified 2.72 million biallelic variants (2365994 SNPs + 354732 indels) that were used in downstream population genomic analyses (Supplementary Data 1).

### Population structure suggests western and eastern glacial refugia in Europe

We first explored the connection between population genetic structure and the geographic distribution of woodland strawberry across Europe (Fig. 1A) through principal component (PC), phylogenetic and admixture (introgression) analyses. The first two PCs of genetic variation accounted for 12.9% and 5.5% of the variance and revealed two main branches corresponding to the two major clusters in maximum likelihood phylogenetic analysis of SNP data (Fig. 1B–D, Supplementary Fig. 1A–F): western European samples from the Mediterranean to northwestern Norway spread along the left branch, while more easterly samples from Romania to northeastern Norway scattered along the right branch (Fig. 1B, Supplementary Fig. 1A–C). A PC analysis on an additional dataset placed samples from central Sweden within the eastern group, together with Finnish samples collected across the country (Supplementary Fig. 2). A Northeast-Southwest PCA cline is particularly visible in inverted form on the left side of

Fig. 1B. This strong geographic pattern reflects isolation by distance (IBD) phenomena[30], as exemplified by humans[31], whereby stepwise fixation of SNPs via random genetic drift occurred in postglacially migrating populations with restricted gene flow (Fig. 1E, Supplementary Fig. 3).

The samples from the southern European peninsulas, Iberia (Spain and Portugal), Apennine Peninsula (northern Italy), and the Balkans (Croatia), formed distinct groups along the western branch, with the Croatian samples clustering closest to the eastern branch. The eastern populations from Kåfjord and Alta—neighboring fjords in northeastern Norway—were the most clearly separated along the second PC. These samples were also clearly separated from Finnish samples previously collected across the country, except for the two northernmost Finnish individuals which grouped close to the Alta samples[32]. Leading-edge populations from Alta and Kåfjord also formed distinct groups in the SNP phylogeny and showed high genetic differentiation with an $F_{ST}$ value of 0.77 (Fig. 1C, Supplementary Table 1), suggesting strong bottlenecks during past colonization and/or isolation in distinct refugia during the Last Glacial Maximum (LGM). This refugial hypothesis is further supported by a markedly weakened IBD pattern when these populations are included (Supplementary Fig. 3B), indicating that their divergence from other regions exceeds expectations under a simple postglacial colonization model.

The geographic contact zone between the western ($N = 107$) and the eastern ($N = 92$) European samples, as robustly inferred by the SNP phylogeny (Fig. 1C, Supplementary Fig. 1E, F, Supplementary Data 2), extended from northern Norway through central and southern Norway and into central Europe, reaching Croatia in southern Europe (Fig. 1A). To examine the geographic distribution of western and eastern ancestries in greater detail, we conducted an ADMIXTURE analysis[33] focusing here on the two major ancestry components ($K = 2$). Samples originating from near the contact zone, for example in Croatia, Germany and central Norway, displayed both eastern and western ancestries in their genomes (Fig. 1D, Supplementary Data 3, Supplementary Fig. 1G). In contrast, samples from Finland, Russia, Alta, and Kåfjord carried exclusively eastern ancestry, whereas Iberian samples were solely of western ancestry.

The contact zone stretching from the Adriatic Sea to the Baltic Sea, which separates western and eastern woodland strawberry, also delineates the geographic ranges of two subspecies of the house mouse (*Mus musculus domesticus* vs. *M. m. musculus*)[34] and the hedgehog (*Erinaceus europaeus* vs. *E. roumanicus*)[35], as well as two admixture groups of silver birch (*Betula pendula*)[15] and Chalk-hill blue butterfly (*Polyommatus coridon*)[36]. This collective demarcation suggests that western and eastern European populations of various organisms were isolated in distinct refugia during glacial periods[2,37]. The observed boundary aligns with a primary contact zone in central Europe, where diverse biota experienced secondary contact during postglacial range expansions[2,38]. Compared with other species in which east-west clustering typically covers ~8–10° of latitude, the division of woodland strawberry into distinct genetic clusters spans a much broader latitudinal range, extending from 45°N in Croatia to 70°N in northern Norway.

The populations follow the boundary between the temperate oceanic and humid continental climate zones in western and eastern Europe, respectively, which differ for several bioclimatic variables, particularly temperature seasonality[39,40]. Owing to their positions relative to the North Atlantic, the climatic differences likely persisted throughout GI cycles[41,42], suggesting that they contributed to genomic differentiation in woodland strawberry (Fig. 1D, F, Supplementary Fig. 4, 5) and other species. To test whether the eastern admixture proportion ($K = 2$) of the genome was significantly associated with 19 bioclimatic variables, we calculated correlations across all samples (Supplementary Data 3). The variables associated with temperature seasonality (Bio4, $r = 0.758$, a Bonferroni corrected $p = 1.21 \times 10^{-34}$ and Bio3, $r = -0.75$, $p = 1.29 \times 10^{-33}$) and winter temperature (Bio6, $r = -0.729$, $p = 7.93 \times 10^{-31}$ and Bio11, $r = -0.691$, $p = 1.90 \times 10^{-29}$) showed the strongest correlations with the eastern admixture proportion (Fig. 1F, Supplementary Fig. 4) and flowering time in the same plant material[43]. Because these bioclimatic variables have been identified as key drivers of adaptation to cold climates in many perennial

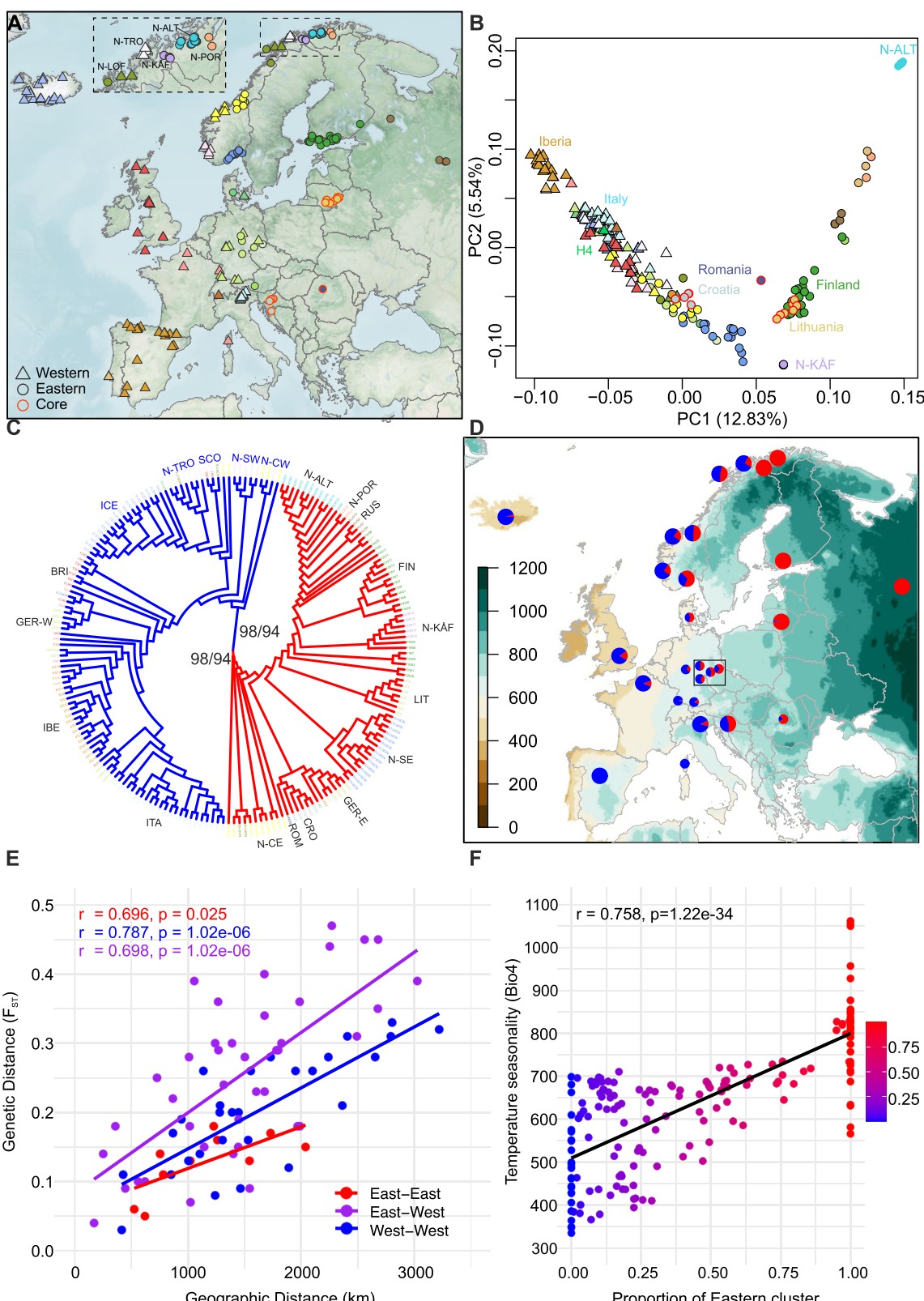

taxa[44], they likely contributed to adaptive genetic clustering in woodland strawberry.

## Habitat fragmentation increases with latitude

To explore the present status of woodland strawberry populations, we estimated the current $N_E$ of each sample by analyzing their most recent coalescence rates[45,46], and assessed the extent of inbreeding by calculating relative inbreeding coefficients based on runs of homozygosity ($F_{ROH}$) across samples[47].

The $N_E$ values of eastern European samples from Lithuania, Croatia and Romania were 5- to 10-fold higher than the median $N_E$ values in other regions (Supplementary Fig. 6A). These samples were also located in the

**Fig. 1 | Collection sites and population structure of European woodland strawberry. A** Collection sites of sequenced woodland strawberry accessions. Western populations are indicated by triangles and eastern by circles; the division is based on the maximum likelihood phylogeny shown in (**C**). Symbols with red outlines indicate core populations with large effective population sizes. **B** Principal component (PC) analysis on the genomic data of woodland strawberry accessions shown in (**A**) using the same symbols. H4 = Hawaii-4 reference accession[87]. **C** SNP phylogeny of European samples. The inner blue and red branches represent western and eastern samples, respectively. Bootstrap support values (*SH-aLRT/ultrafast*) for these branches are shown. For visual clarity, the names of sampled regions within the northern sub-branch of the western branch are colored blue. Outer branch colors correspond to colors in (**A**, **B**). **D** Average ancestry proportions derived from ADMIXTURE analysis (*K* = 2), plotted for regional sample pools, from different countries/areas of countries or specific samples (large and small pies, respectively)

on a background gradient map of temperature seasonality (BIO4; standard deviation × 100). Blue = western ancestry, Red = eastern ancestry. German samples are shown with small symbols to highlight the gradient in the west-east admixture. The black rectangle shows the contact zone. **E** Isolation by distance (IBD) correlations within western and eastern and between western and eastern regions, excluding northern Norwegian outlier regions Alta and Kåfjord. **F** Correlation between the eastern admixture proportion of the genome and temperature seasonality (BIO4) of sample collection sites. Abbreviations: BRI Britain, CRO Croatia, FIN Finland, GER-W Western Germany, GER-E Eastern-Germany, IBE Iberia, ICE Iceland, ITA Italy, LIT Lithuania, N-ALT Norway-Alta, N-CE Norway-central-eastern, N-CW Norway-central-western, N-KÅF Norway-Kåfjord, N-LOF Norway-Lofotes, N-POR Norway-Porsanger, N-SE Norway-southeastern, N-SW Norway-southwestern, N-TRO Norway-Tromsø, ROM Romania, RUS Russia, SCO Scotland.

core of the PCA plot; we hereafter term these "core populations" (Fig. 1A, B). The lowest $N_E$ values were observed at the range edges, specifically in the Iberian Peninsula, Iceland, and northern Norway (Supplementary Fig. 6A, 7). Moreover, strong negative latitudinal correlations in regional $N_E$ values were found in both western and eastern Europe (Fig. 2A, B).

Based on $F_{ROH}$ values, the core populations were the least-inbred, with $F_{ROH}$ values close to 0.15 (Fig. 2C, D, Supplementary Fig. 6B). They were also the only populations that were close to Hardy-Weinberg equilibrium based on $F_{IS}$ values close to 0 (Supplementary Fig. 8, Supplementary Data 3; see "Methods"). This indicates that populations in these regions exhibit panmictic characteristics with low levels of habitat fragmentation and inbreeding. Furthermore, a positive latitudinal correlation was observed in $F_{ROH}$, particularly in the western genetic cluster, with $F_{ROH}$ values ranging from 0.5 in northeastern Iberia to 0.75 in Iceland and Tromsø in northwestern Norway (Fig. 2C, D). Taken together, the strong latitudinal correlations in $N_E$ and $F_{ROH}$ suggest pronounced founder effects in northern populations and increasing inbreeding towards the north, consistent with postglacial range expansion dynamics[48,49]. We found no latitudinal correlation in region-wide expected heterozygosity π (Supplementary Fig. 9), which can be inflated by among-subpopulation variation suggesting that samples became increasingly differentiated towards the north due to habitat fragmentation. A recent study on core-periphery dynamics showed that animal-dispersed plant species, such as woodland strawberry, are particularly sensitive to habitat fragmentation[29]. In southwestern Europe, samples from the Pyrenees (NE-Iberia) harbored low π (Supplementary Fig. 9A), reflecting historically small $N_E$, in contrast to relatively high current $N_E$ (Fig. 2A). This suggests that current $N_E$ inferred from the most recent coalescence rate more accurately reflects present conditions than the widely used π measure, as demonstrated in humans[45,50].

**Southern refugia of woodland strawberry span four ice ages**

To explore the demographic history of woodland strawberry, we analyzed population divergence and calculated time-dependent coalescence rates from whole, diploid genomes using the Multiple Sequentially Markovian Coalescent 2 (MSMC2) method[45], which does not require a predefined demographic model. We then applied an isolation-migration (IM) model to these rates[46] to infer stepwise changes in historical $N_E$ and migration rate between populations, providing continuous estimates of population divergence through time. We assumed a 2-year generation time, as previously used for other perennial herb species[51,52] and tested, in multiple datasets, both the experimentally determined mutation rate of *Arabidopsis thaliana* ($7.1 \times 10^{-9}$ mutations per nucleotide per generation[53]; see also [https://www.nature.com/articles/s41588-019-0442-7]) and the evolutionary mutation rate estimated for the *Fragaria* genus[13]. This calibration showed that the *Arabidopsis* mutation rate provided a more accurate alignment of demographic events with the timing of past GI cycles[1] than the evolutionary mutation rate estimated for the whole *Fragaria* genus (Supplementary Figs. 10–14).

Given the strong genetic clustering between western and eastern populations, we investigated the timing of their initial genomic divergence

($M < 0.999$ and $M < 0.99$) (Supplementary Fig. 15). Among the sample pairs between western (Iberian) and eastern (Romanian/Lithuanian) populations, initial divergences consistently dated to 450–330 ka (Supplementary Fig. 15A), representing the maximum temporal resolution achieved in this study. This period spans the transition from marine isotope stage 11 (MIS 11) to the MIS 10 glacial period, suggesting that western and eastern woodland strawberry evolutionary lineages were established at least 330 ka, spanning four ice ages. Several independent sample pairs from the Italian Alps and the Iberian peninsulas also began to diverge ~330 ka ago (Supplementary Fig. 15A, Supplementary Data 4), ~100 ka earlier than alpine rockcress populations in the same regions[54]. These distinct ancestries between the peninsulas were further supported by the ADMIXTURE analysis (*K* = 6 and *K* = 8; Supplementary Fig. 1G, H). Drawing parallels with other species, mitochondrial divergence analyses of brown bear (*Ursus arctos*), grasshopper (*Chorthippus parallelus*) and tawny owl (*Strix aluco*), and pollen records of beech (*Fagus sylvatica* L.) suggest that these species likely occupied refugia in the western and eastern peninsulas around the same time[2,55,56].

**Core and peripheral populations show contrasting demographic patterns**

We inferred the demographic history of woodland strawberry using haplotype data, which provides good temporal resolution even with few samples. A reduction in historical $N_E$ and an excess of isolation events were observed during the LGM and PGP across the full dataset, while both $N_E$ and migration rates showed substantial variation among samples (Supplementary Fig. 10, 16A). Further analysis using samples that best fulfilled the expectations of MSMC2/MSMC-IM methods[45,46] revealed common contrasting demographic patterns that aligned with the chronological sequence of serial MIS (Supplementary Note 1, 2, Supplementary Fig. 17): a "core pattern" characterized by stable $N_E$ and migration rate across multiple GI cycles until the LGM (Fig. 3A, B, Supplementary Fig. 16B, C), and a "peripheral pattern" defined by recurrent isolation events and bottlenecks during glaciations followed by recontacts in subsequent interglacial periods (Fig. 3C–H). The stable core pattern was most frequently detected when haplotypes were drawn from two eastern core populations with high $N_E$ from Croatia, Romania, or Lithuania, while other haplotype combinations more rarely showed this pattern (Supplementary Data 4, Supplementary Fig. 17C–E). The climate sensitive peripheral pattern was typically observed in sample pairs from different peninsulas as well as between northern and southern samples (Supplementary Figs. 17C, E, 18) possibly indicating the existence of separate northern microrefugia during glacial periods. In general, these patterns support previous studies[13,15], showing that species with larger $N_E$ are more resilient to glacial periods.

In the core pattern, found in 14.2% of MSMC-IM runs (Supplementary Fig. 17A), $N_E$ consistently exceeded 20,000 throughout GI cycles (Fig. 3A, Supplementary Fig. 19A–D). Only a slight decrease in $N_E$ was observed towards the PGP, which is recognized as a stronger glacial period than the earlier MIS 10 and MIS 8 stages[1]. This observation, together with the stable migration rates observed among core populations across GI cycles (Fig. 3B,

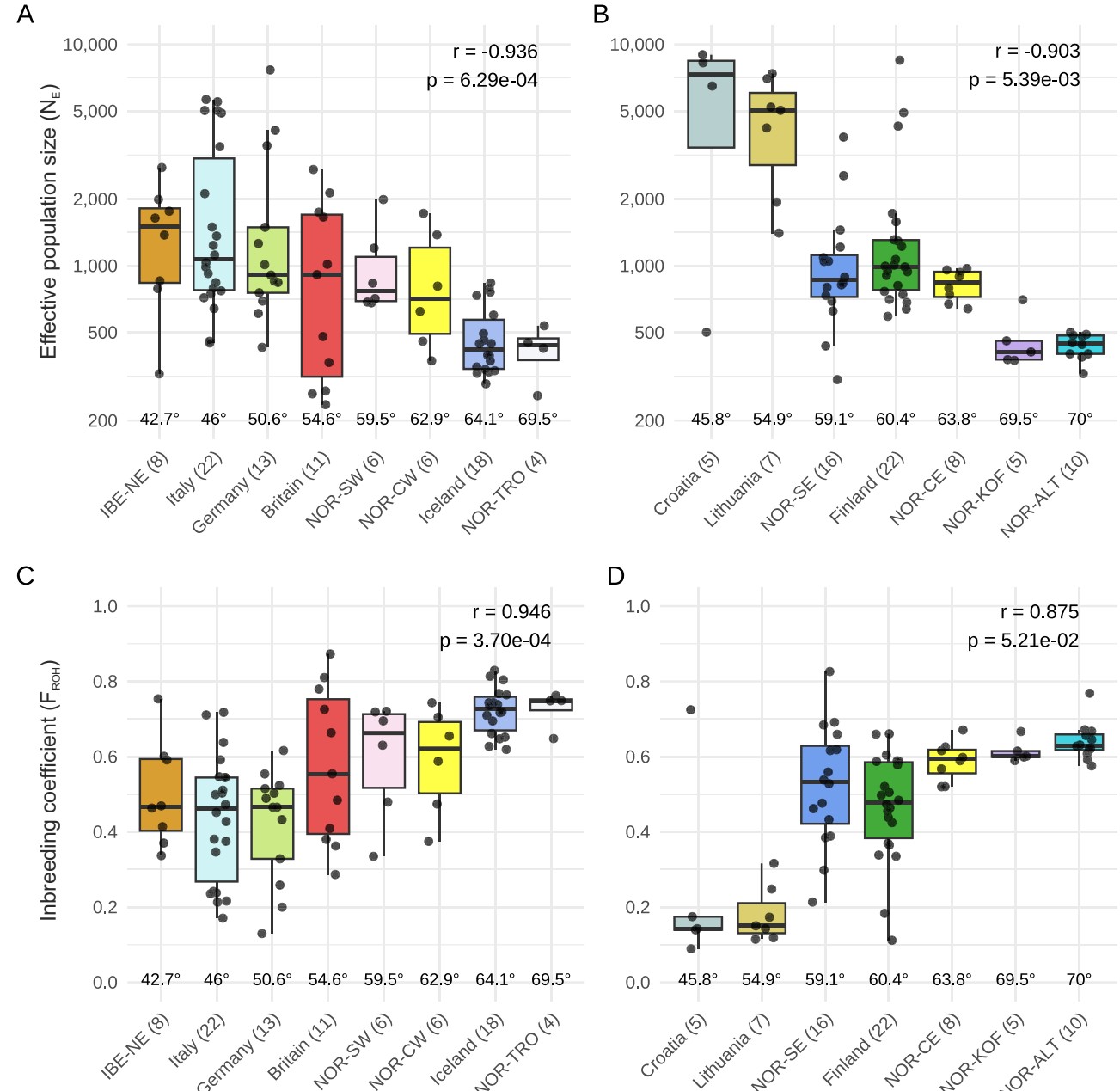

**Fig. 2 | Latitudinal patterns of effective population sizes and inbreeding coefficients. A, B** Latitudinal correlations of median (most typical sample per region) effective population sizes ($N_E$) in western (**A**) and eastern (**B**) European samples. **C, D** Latitudinal correlations of median inbreeding coefficients ($F_{ROH}$) of regional sample sets from western (**C**) and eastern (**D**) Europe. The number of samples from each region is indicated in the legends after the name of each region. The Pearson correlation coefficient (*r*), calculated across regional medians, with statistical significance indicated by the *p* value.

Supplementary Fig. 16B, C), suggests that ancestral haplotypes with the core pattern originated from a large stable ancestral population that remained intact during glacial periods. Bootstrap analyses supported this pattern, indicating that isolation events and bottlenecks were rare between MIS 10 and the LGM (Supplementary Figs. 16C, 19). The frequent emergence of the core pattern in analyses of Croatian and Romanian haplotype pairs (Supplementary Fig. 17E) suggests that a common ancestral population of woodland strawberry may have persisted in southeastern Europe, possibly within the Balkan Peninsula, a region widely recognized as a major refugium for numerous species during the Pleistocene[2,37,57,58]. An alternative hypothesis that warrants further study is that woodland strawberry maintained a large stable population in southwestern Asia, for example in the Caucasus region, which may have acted as a recurrent source of migrants into Europe[59]. Although broader sampling of southern European habitats is

needed to draw firm conclusions about the geographic origin of core populations, our results (Supplementary Fig. 6, 7) align well with earlier studies on e.g., alpine rockcress[20], primrose (*Primula vulgaris* Huds.)[60], gray wolf (*Canis lupus*)[61], and dunnock (*Prunella modularis*)[62], all of which show their largest population sizes in the Balkans. This pattern likely reflects the region's role as a major source area for postglacial colonization, consistent with Hewitt's refugial paradigm[2,57,63,64].

Peripheral patterns, characterized by a cessation of migration and/or a bottleneck between MIS 10 and the LGM, were detected in more than 56% of MSMC-IM runs (Supplementary Fig. 17A, Supplementary Note 1). A strong peripheral pattern, observed in 19% of runs (Peripheral-1; Supplementary Fig. 17A, B), showed a high sensitivity to consecutive marine isotope stages, with migration rates fluctuating across multiple GI cycles and strong bottlenecks associated with either the PGP or MIS 8 glaciations

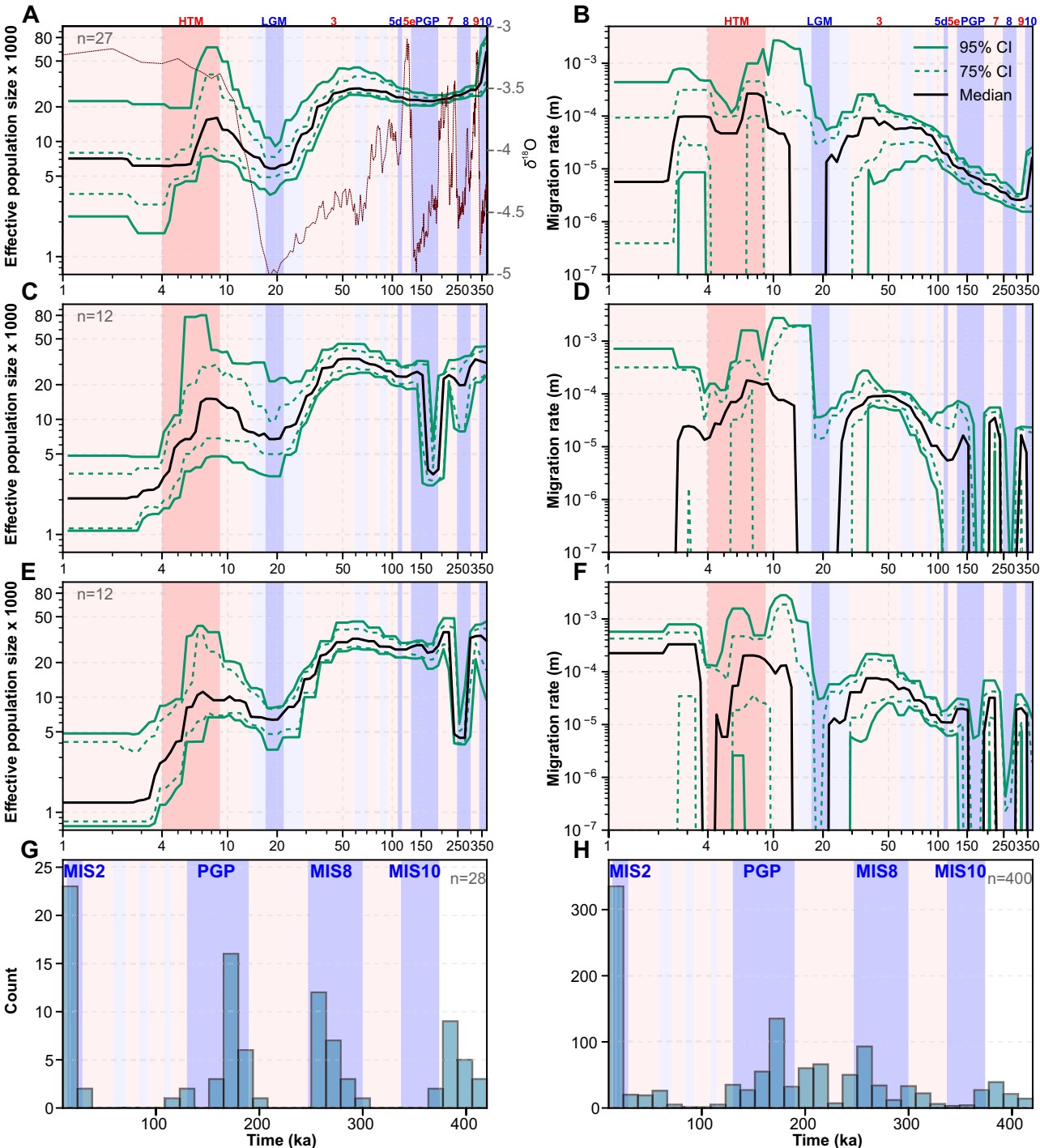

**Fig. 3 | Demographic history of European woodland strawberry.** Effective population sizes ($N_E$) and historical migration rates (m) through time in the core pattern (**A, B**), and in the strong peripheral patterns showing strong bottleneckcs during the PGP (**C, D**) or MIS 8 (**E, F**). To estimate confidence intervals for demographic trajectories (**A–F**), replicate MSMC-IM curves were summarized by computing empirical percentiles across biological replicates on a common time grid ($N = 80$ time points). For each time point, we calculated the 50th percentile (median) and the 12.5–87.5% and 2.5–97.5% percentile envelopes. **G, H** Isolation event midpoint ($m < 1 \times 10^{-7}$) frequency in the peripheral pattern through time using

biological replicates with the lowest $F_{ROH}$-values from each region (**G**) and across all peripheral bootstrap replicates (**H**). X-axis values represent thousands of years. Glacial and interglacial periods are shown with background colors, and blue and red numbers indicate specific marine isotope stages (MIS) and substages (a letter after the number). Purple curve at A represents inverse benthic δ18O records from Lisiecki and Raymo (2005)[1] and are used as a proxy for historical temperature. HTM Holocene thermal maximum (9–4 ka), LGM Last glacial maximum (22–17 ka), and PGP Penultimate Glacial Period (190–130 ka).

(Fig. 3C–F, Supplementary Figs. 12–14). Consistent with whole-genome results, the bootstrap analysis (400 runs) also revealed increased isolation frequencies during the prolonged PGP (~60 ka) and MIS 8 (~57 ka), and possibly during MIS 10 with less accurate timing due to lower MSMC-IM

resolution in the distant past (Fig. 3G, H, Supplementary Data 4, Supplementary Fig. 18). Temporal deviations between bootstrap replicates and whole-genome inferences may arise from heterogeneous selection across the genome, including genetic hitchhiking[65–67] and background selection[68,69].

These effects are expected to be the strongest in genomic regions of low recombination and in predominantly selfing or asexual populations[68], where reduced effective recombination magnifies the impact of linked selection and increases variability in evolutionary rates across the genome.

## The LGM caused a strong bottleneck in all European woodland strawberry populations

After the PGP, $N_E$ values increased in all samples towards the beginning of MIS 3 interstadial (59–29 ka), a period when plant populations flourished across Europe[70–72]. This was followed by accelerated declines in $N_E$ towards the onset of the LGM (Fig. 3A, C, E, Supplementary Figs. 10–14A, B, 18A–D, 19A–D, S20A, B) consistent with patterns reported in other species[12–14]. Concurrently, gene flow between populations abruptly ceased in most sample pairs (Fig. 3B, D, F–H, Supplementary Figs. 11–14C, D, 18E–H, 19E–H, 20C, D, 21). Unlike during the three preceding glacial periods, core populations also experienced severe bottlenecks during the LGM (Fig. 3A, Supplementary Fig. 19A–D). The most pronounced declines in $N_E$ during MIS 2 were observed in populations currently located at the range edges, including samples from Alta (Supplementary Fig. 20A, $p = 5.05 \times 10^{-11}$), Kåfjord (Supplementary Fig. 20B, $p = 2.96 \times 10^{-5}$), Finland ($p = 1.90 \times 10^{-5}$), and the Iberian Peninsula ($p = 3.87 \times 10^{-5}$). These declines were notably significant when compared with $N_E$ in e.g. Croatia or Italy during MIS 2 (Supplementary Data 4), suggesting that northern populations survived the LGM either in peripheral areas of major southern refugia or in distinct microrefugia, as suggested for other species[73,74].

During the transition to the Holocene, $N_E$ values began to increase, reaching their highest post-LGM levels during the first half of the Holocene Thermal Maximum (HTM, ca 9–4 ka; Fig. 3A, C, E, Supplementary Fig. 10–14A, 18A–D, 19A–D, 20A, B), a period broadly documented in paleoclimatic records, particularly from central and northern Europe[41,75–77]. Following peak $N_E$ during the HTM, population sizes began to decrease, with stronger declines towards northern latitudes (Supplementary Fig. 22, Supplementary Data 4). In these regions, climatic events can amplify temperature changes several-fold relative to the global average, as has been documented over the past four decades[78]. This could potentially contribute to strong founder effects. Current $N_E$ values are markedly lower than those during earlier interglacial periods (Supplementary Fig. 10A), a trend also observed in other plant species[13,19,54,79]. This indicates a higher degree of habitat fragmentation during the late Holocene compared with earlier interglacial periods, coinciding with the onset of deforestation in Europe around 5 ka[80]. Consistent with earlier glacial periods, Croatian and Romanian samples showed the highest resilience to this latest habitat fragmentation (Supplementary Fig. 22, Supplementary Data 4), although this fragmentation ultimately led to their divergence (see below).

## Northern Europe was colonized from both sides of the continent in several waves

To investigate continent-wide postglacial colonization patterns, we estimated split times ($M < 0.5$) between regions. Split times of peripheral populations were calculated from the largest current source populations (e.g., from Lithuania and Italy) based on the assumption that these populations best represent the ancestral sources of colonizing migrants (Supplementary Fig. 23, Supplementary Data 4). Although early signals of divergence were detected between putative refugia (Supplementary Fig. 15), most of the genome remained undiverged (median cumulative migration probability, $M > 0.5$) until the LGM, when gene flow ceased between populations (Fig. 3B, D, F–H, Supplementary Figs. 11–14C, D, 16, 18E–H, 19E–H, 21), and splits between several western (Iberian) and eastern (Romanian or Lithuanian) samples were observed (Fig. 4A, Supplementary Fig. 24, Supplementary Data 4, 5). These splits coincided with the separation of predicted ecological niches for western and eastern ecotypes of wild grapevine during the LGM[14], further supporting the role of the LGM in driving population divergence. Additionally, the decline in effective population sizes towards the LGM likely contributed to the separation of western and eastern ancestries in both woodland strawberry (Supplementary

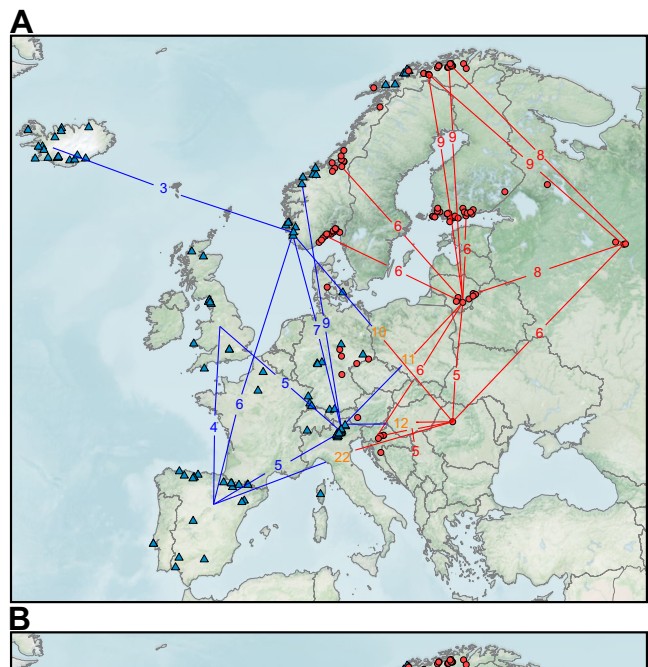

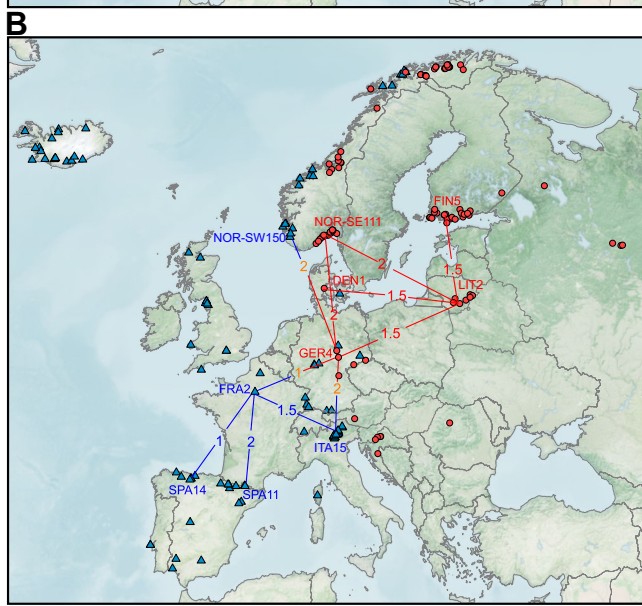

**Fig. 4 | Divergence in European woodland strawberries. A** Median split times in thousands of years (ka) between the regional samples during the postglacial northward migration of woodland strawberry. **B** Split times between the largest regional populations from northern Iberia to southern Scandinavia. Blue and red lines/numbers indicate divergence within the western and eastern genetic cluster, respectively, while orange numbers show the split time (ka) between western and eastern samples.

Fig. 10A) and grapevine[14]. Further analyses showed that the median split time between western and eastern sample pairs (9.8 ka) was significantly earlier than the median split times within western (5.6 ka, $p = 4.05 \times 10^{-12}$) and eastern (5.9 ka, $p = 2.02 \times 10^{-13}$) pairs (Supplementary Fig. 24B, Supplementary Data 5), supporting the idea that western and eastern Europe were mainly colonized by western (i.e., Iberian or Italian) and eastern ancestry, respectively, during the Holocene (Fig. 4A). This pattern is further supported by higher genetic differentiation between eastern and western regions than within each group (Supplementary Fig. 2A).

In eastern Europe, the Lithuanian population split from Croatia and Romania 6–5 ka ago, around the same time that Croatia and Romania diverged from each other, with the narrow variance in split times (Supplementary Fig. 24C, D) indicating that panmictic populations faced a rapid habitat fragmentation. Peripheral populations from Alta, Kåfjord and

Russia, along with some samples from Finland and southeastern Norway diverged from Lithuanian samples during the first half of the Holocene, primarily during the early HTM (9–6 ka) (Fig. 4A, Supplementary Fig. 24A–C), a period of amplified warming in the European Arctic[75,81,82], while the rest of the eastern samples split during the late HTM (6–4 ka) (Supplementary Fig. 24A). The split times of the Alta and Kåfjord samples from northwestern Russian samples were within a similar range as from Lithuania (Fig. 4A), suggesting that a large Lithuanian source population (Supplementary Data 4), together with Russian ancestry (Supplementary Fig. 1G) contributed to Arctic colonization. The early split of the Alta and Kåfjord samples from the Lithuanian lineage was followed by a severe bottleneck with a transient $N_E$ decline of 98.3% and 96.0%, respectively, relative to peak HTM levels. This suggests that the most recent colonization wave of eastern woodland strawberry ancestry reached the Arctic region during that time. However, several lines of evidence indicate that a substantial fraction of Alta and Kåfjord genomes represent relicts from earlier glacial cycles. They harbor very low nucleotide diversity ($\pi_{NOR-ALT}$ = 0.00022, $\pi_{NOR-KÅF}$ = 0.00026, Supplementary Fig. 9B) compared with other regions (e.g., $\pi_{Croatia}$ = 0.0017), they were isolated from the Lithuanian population during the last glacial cycle prior to the LGM (Supplementary Fig. 20C, D), admixture analysis with an optimal number of $K = 8$ revealed their unique ancestries (Supplementary Fig. 1G, H), and their genetic differentiation from other regions exceeded levels expected from post-glacial colonization history alone (Fig. 1E vs. Supplementary Fig. 3B). As suggested for beech[74], these data support the existence of northern microrefugia for the Alta and Kåfjord populations. These populations may harbor unique adaptations to the Arctic region, including extreme early flowering and early formation of winter leaves[43,83], and are at risk of extinction due to the rapid warming of the Arctic[78].

In western Europe, Britain was colonized by populations of western ancestry (Supplementary Data 5, Supplementary Fig. 24). Although mean split times from Iberian (3.8 ka) and Italian (6 ka) samples did not differ significantly (Fig. 4A, Supplementary Fig. 24C), the admixture analysis indicated a predominant contribution of Iberian ancestry to the colonization of Britain (Supplementary Fig. 1G). Germany, located at the boundary between the eastern and western genetic clusters, was primarily colonized from Italy, as indicated by significantly shorter split times from Italy compared with Iberian ($p = 0.0026$), Croatian ($p = 0.0378$), or Romanian ($p = 0.0061$) samples (Supplementary Data 5, Supplementary Fig. 24C).

Norwegian coastal regions (NOR-SW, NOR-CW, NOR-CE) were likely colonized from both sides of Europe, as no significant differences were detected in split times between western (Iberian or Italian) and eastern (Lithuania or Romania) ancestries (Supplementary Data 5, Supplementary Fig. 24C). While most of these samples split during the HTM, several northwestern individuals from Scotland (UK1 and UK6), SW-Norway and CW-Norway, Iceland, and Tromsø diverged much earlier, likely before the onset of the Holocene (Supplementary Data 4; Supplementary Fig. 24C). Their effective population sizes were already much lower by MIS 2 compared with those of populations that diverged during the HTM. In addition, these samples clustered within the northern sub-branch of the western lineage in the SNP-based phylogenetic tree (Fig. 1C). Together, these findings suggest that substantial portions of these genomes originated from a northwestern glacial refugium, similar to what has been proposed for lyrate rockcress[19] and for the northeastern Alta and Kåfjord populations in this study. Finally, several Icelandic subpopulations diverged from southwestern Norway between 3 and 1.3 ka (Fig. 4A, Supplementary Fig. 24C, Supplementary Data 4), indicating that the most recent colonization of woodland strawberry in Europe occurred in Iceland and was likely influenced by human activity.

Further analysis of samples with the largest current $N_E$ from Iberia (ES14 and ES11), France (FRA2), Italy (e.g., IT15 and IT2), Germany (GER4), Lithuania (LIT1 and LIT2), southeastern Norway (NOR-SE111 and NOR-SE120) and Finland (FIN5 and FIN13) revealed recent split times from their closest neighbors, comparable to those among Lithuanian subpopulations (Fig. 4B, Supplementary Data 4). This implies that the largest populations, of which several were admixed, formed a connected population chain extending from northern Spain and Italy to the southern parts of the Nordic countries, with gene flow between eastern and western Europe occurring primarily through Germany (Fig. 4B). In the face of ongoing climate change and accelerated warming at higher latitudes[78], this population chain is expected to expand northward. However, forest habitat fragmentation, which began ~5000 years ago[80], now poses a major threat, as most populations are small and vulnerable to extinction. While the current migration network continues to maintain genetic diversity at a broad scale, it remains uncertain how long this connectivity can persist under ongoing environmental change. Notably, the large Croatian and Romanian populations, although not part of the main migration chain, retain substantial genetic diversity that is likely to play an important role in the species' survival during future GI cycles.

## Concluding remarks

Here, we report a clear division of European woodland strawberry into western and eastern genetic clusters, closely associated with the temperature seasonality of their habitats. These groups were isolated in western and eastern refugia during late-Pleistocene glaciations and subsequently came into secondary contact after northward range expansion during interglacial periods. Consistent with ecological theory[84], peripheral populations were more susceptible to isolation and bottlenecks, while current core populations were part of the stable source that persisted through multiple Pleistocene glacial cycles. This source population served as a crucial reservoir of genetic diversity, as evidenced by admixture between core and peripheral populations during interglacial range expansions. Our results suggest that, from a conservation perspective, preserving current core populations and other selected populations with high $N_E$ is essential for maintaining the long term genetic diversity of the species. From an evolutionary perspective, the ability to trace ancestral haplotypes to distinct historical periods, particularly when future work accounts for variable evolutionary rates across genomic regions (i.e., haplotype-specific nucleotide divergence, $K$, between species), has the potential to provide new insights into genome evolution and climatic adaptation in temperate flora.

## Methods
### Samples, DNA extraction and sequencing
Wild accessions of woodland strawberry were collected as clones or seeds, or obtained as clones from existing collections at Hansabred, Germany and IFAPA, Spain. Single seedlings were raised from seeds collected from each location, and together with clonal plant materials, the collection (Fig. 1A, Supplementary Data 1), covering majority of the European distribution of the species[21], was maintained in a greenhouse at the University of Helsinki. For DNA extraction, young, folded leaves were collected and frozen in liquid nitrogen followed by DNA extraction using CTAB-protocol[85]. Sequencing libraries were constructed using Nextera DNA Flex (now DNA prep) protocol according to the manufacturer's instructions (Illumina). Sequencing was performed with an Illumina NextSeq 500 machine using a paired end sequencing runs (170 bp and 140 bp, or 151 bp and 151 bp).

### Trimming, mapping and initial variant calling
Raw Illumina reads were trimmed and paired with Cutadapt (DOI:10.14806/ej. 17.1. 200) using -q 25 and -m 30 parameters. Trimmed reads were mapped with BWA-MEM[86] against the *F. vesca* "Hawaii-4" reference genome v.4.0[87] with -M option. Duplications were marked with the Picard tool MarkDuplicates. GATK v 3.7. HaplotypeCaller-function was used to call variants in single samples, and then, variants were jointly called across all samples using the GenotypeGVCFs-function. Indels were filtered with following parameters: QD < 2.0||FS > 200.0||SOR > 10.0 || InbreedingCoeff < −0.8 || ReadPosRankSum < −20.0, and SNPs: QD < 2.0||FS > 60.0|| SOR > 4.0|| MQ < 40.0 || MQRankSum < −12.5 || ReadPosRankSum < −8.0.

## SNP panel

Following use of the GATK pipeline, SNPs and indels closer than 10 bp from indels were removed with bcftools[88], which resulted in 3,479,750 variants (2,898,156 SNPs and 581,594 indels) across samples. On average, 5.3% of missing data per polymorphic site was found. Several additional filtering steps were performed to remove low quality SNPs using vcftools v.0.1.16[89] before imputation and phasing. Initially, we included only those biallelic sites (minor allele frequency > 0.001, max-alleles 2) that had a maximum of 10% missing data and a mean coverage of less than 29, approximately twice the average coverage. Excessively heterozygous sites, likely erroneous in this species in which most populations exhibit clear signs of inbreeding, were removed from all samples if they deviated from Hardy-Weinberg equilibrium ($p < 0.01$) in any regional set of samples. Excessively homozygous sites were not removed because of the self-compatible nature of *F. vesca*. After additional filtering, 2,720,726 biallelic variants (2365994 SNPs + 354732 indels) remained. Imputation was conducted using BEAGLE v4.1 (Browning et al. 2014)[90] with default settings. Phasing was conducted for all samples simultaneously using SHAPEIT v2 software[91]. Phase informative reads (PIR) were utilized in phasing with the extractPIRs-tool included in the SHAPEIT-software. The following parameters were used in phasing: −rho 0.001 --states-random 200 --window 0.5 --burn 10 --prune 10 --main 50. Variants were annotated with SnpEff-software v.4.3[92] prior to extracting 4-fold degenerate sites from synonymous sites.

## Population structure

Principal component analysis (PCA) was conducted with SNPrelate software v.1.40.0[93] using polymorphic 4-fold degenerate sites (41,179 SNPs), non-synonymous sites (84,233 SNPs), and genome-wide using linkage disequilibrium pruned sites ($R^2 < 0.2$ in 20 kb region, 40,827 SNPs) with a minor allele frequency greater than 0.01. ADMIXTURE software v.1.3.0[33] was used to determine admixture proportions of genetic clusters for each sample. Cross-validation error (CVE) was the lowest at $K = 8$ (Supplementary Fig. 1H), which was selected to represent population structure in the admixture data set. However, since CVE was reduced strongly already before $K = 8$, possibly due to large amount of hybrid samples in the dataset, we also included $K = 4$ and $K = 6$ graphs for comparison. To visualize eastern and western clusters across accessions, $K = 2$ was used. A phylogenetic tree was inferred using *IQ-TREE v.1.6.12*[94] under maximum likelihood for each of three genomic datasets: *four-fold degenerate, missense*, and *LD-pruned SNPs*. The best-fit substitution model for each dataset was selected using the *Bayesian Information Criterion (BIC)*. Models included $TVM + F + ASC + R6$ (four-fold), $GTR + F + ASC + R7$ (missense), and $GTR + F + ASC + R5$ (LD-pruned). All analyses incorporated *Lewis ascertainment bias correction*[95] for SNP data and used *rate heterogeneity across sites* modeled with discrete Gamma categories. Branch support was assessed using both *SH-aLRT* and *ultrafast bootstrap (UFBoot)* with 1000 replicates. Genetic differentiation between regions was estimated by weighted $F_{ST}$[96] for regional pairs of samples ($N = 5$). In Iberia, we used samples from northeastern Iberia (NE-Iberia) for $F_{ST}$ analysis due to enough samples ($N = 8$) from this region, while other Iberian samples were scattered across the peninsula and primarily represented edge-of-range populations.

## Isolation by distance

IBD was studied based on correlation of geographic distance (kilometers) and genetic distance ($F_{ST}$) between the regions (Supplementary Table 1). To calculate geographic distances between regions, the mean latitude and longitude were calculated for each region based on the GPS coordinates of its samples. These mean coordinates were then used to estimate the geographic distances between the regions. Pairwise correlations were calculated separately for eastern (>50% eastern samples) and western regions (>50% western samples) and between eastern and western regions using the cor.test function implemented in R version 4.3.2.

## Population structure-environment correlations

Climatic data for 19 bioclimatic variables of plant collection sites were extracted from the WorldClim 2.0 dataset at 30-s resolution based on sample coordinates (Supplementary Data 3). To explore association of these bioclimatic variables with the east-west population structure, their correlations with the eastern admixture proportion were calculated. A few samples missing bioclimatic data due to inaccurate coordinates were excluded from the analysis.

## Inbreeding coefficients

Inbreeding coefficients were calculated in two different ways. $F_{ROH}$ were calculated based on lengths of runs of homozygosity (ROH) for all samples using PLINK v.1.9.0-b.7.7[97]. ROHs were identified for each sample's genome using 50 kb window size with the following parameters: --homozyg-density 100 --homozyg-gap 500 --homozyg-kb 50 --homozyg-snp 35 --homozyg-window-het 1. $F_{ROH}$ was then calculated for each sample based on a proportion of total length of ROHs in the genomes including centromeres (genome length = 219291370 bp). For comparison and to identify regional outlier samples, traditional inbreeding coefficients ($F_{IS}$) for regions were calculated with vcftools[89] for samples in each region or country (Supplementary Fig. 8B, C, Supplementary Data 3) when at least four samples were available from the same geographic region. Note that our particular sampling biases results to some extent. We do not have fully population-level samples; instead, we sampled one individual per population from several separate populations within regions. Thus, we have population structure within the regions (except in Croatia and Lithuania), which increases a proportion of homozygous genotypes within regions, a phenomenon also called the Wahlund Effect[98]. Especially samples from Iberia, Germany and Britain have been collected from large geographic areas around the countries. Despite suboptimal sampling, the observed correlation (Supplementary Fig. 8B) between $N_E$ and $F_{IS}$ across samples suggests that our $F_{IS}$ estimates are also broadly reliable. Two samples from Iberia (ES21 and ES3) and Tromsø (NOR30 and NOR28) were excluded from analyses based on highly negative $F_{IS}$ values compared to other regional values (Supplementary Fig. 8C, Supplementary Data 3), probably caused by recent admixture (Supplementary Fig. 1G). Median $F_{ROH}$ and sample latitudes were calculated from each region to explore correlation between $F_{ROH}$ and latitude across Europe both in western and eastern European samples, based on the clustering of the majority of regional samples (>50%) in the SNP phylogeny.

## Inferring demographic history of populations

Demographic history of populations was inferred from statistically estimated haplotypes. Multiple Sequentially Markovian Coalescent (MSMC)-based method, MSMC2 v.2.1.1 (https://github.com/stschiff/msmc2)[45], was used to estimate coalescence rates within and across pairs of populations. To obtain a time-dependent estimate of migration rate and effective population sizes for each population pair, a continuous Isolation-Migration model was fitted to coalescence rates using MSMC-IM-software[46], where migration between a pair of populations is quantified by piecewise constant migration rate changes between populations. To ensure the highest possible quality of the SNP data for coalescence analyses, as recommended by simulation-based evaluations[99], SNPs were post-filtered with recommended additional filters for each sample (bam-files) and universally for the genome. Specifically, the bamCaller.py script (https://github.com/stschiff/msmc-tools/blob/master/bamCaller.py) was used to produce additional sample-specific masks for low quality SNPs potentially produced by the pipeline. To ensure high-confidence site calls and reduce false positives, particularly important in this species with low levels of heterozygosity (Supplementary Data 3), we used the bamCaller.py script with base quality >20 and mapping quality >20 as input parameters. The script's default settings were used to retain only sites with sequencing depths between 0.5× and 2× of the mean depth of each sample's BAM file. Across all samples, the average minimum and maximum depth thresholds were 9.2 and 36.6, respectively, with only eight samples falling below the minimum depth threshold of six (Supplementary Data 1). In addition, Heng Li's SNPable software (https://lh3lh3.users.sourceforge.net/snpable.shtml) was used to produce universal masks for 35 bp regions (35mers) that were not found uniquely in the *F. vesca* genome[87].

First, the MSMC2 was run using default parameters with 20 iterations to uncover within-population (haplotype pairs 0–1, 2–3) and cross-population (haplotype pairs 0–2, 0–3, 1–2, 1–3) coalescence rates. The -s option was used to skip sites with ambiguous phasing. Since the mode of reproduction and $N_E$ varies between populations, we did not use a fixed recombination rate in our MSMC2 analyses. Instead, population-scaled recombination rates ($\rho$) and mutation rates ($\theta$) were inferred directly from the haplotype data, with the ratio of $\rho$ to $\theta$ expected to be lower in populations with high levels of selfing or asexual reproduction. A low $\rho/\theta$ ratio is usually not problematic for coalescent-based models (except in the case of linked selection biasing demographic modeling), because it improves the detectability of historical recombination events within the mutational landscape, a key prerequisite for reconstructing the Ancestral Recombination Graph[99]. However, because MSMC2 was developed for obligately outcrossing species[45], an assumption not met in this species with a low heterozygosity due to ability to self, which may bias the inference[100], we selected from each region the samples with the lowest inbreeding coefficients ($F_{ROH}$) and/or the highest heterozygosity (to maximize the number of informative recombination events) as our primary dataset ($N = 41$; Supplementary Fig. 25, 26) for demographic modeling. Note that generation time, genomic data filtering[99], and masking of different genomic regions in different samples depending on sequencing quality also influence the timing of inferred events. Then MSMC2-results were analyzed with the MSMC-IM software using default parameters and recommended beta values for regularization of gene flow (1e-8) and $N_E$ (1e-6). Several ancestral $N_E$ values (30,000, 100,000 and 1,000,000) were tested for different sample pairs, but the default ancestral $N_E$ (15,000) was chosen, since it clearly had the highest sensitivity to detect the demographic events beyond the LGM (i.e., PGP). Two haplotypes (single sample) per region were used, thus four haplotypes ($4 \times 7 = 28$ chromosomal haplotypes) in total, in cross-population analyses. Multiple sample pairs between regions were analyzed to confirm the robustness of results (Supplementary Data 4), account for stochastic variation in the data, and were used to assess the timing of initial divergence between populations (Supplementary Fig. 15) and the significance of split times between regional pairs (Supplementary Fig. 24, Supplementary Data 5). In addition, a block-bootstrap approach was used to evaluate how different combinations of genomic regions influence demographic inference, using sample pairs from primary data that represented strong peripheral (Peripheral-1) and core patterns. Resampling was performed with $30 \times 1$ Mb blocks across the seven chromosomes, repeated 100 times with the multihetsep_bootstrap.py script (https://github.com/stschiff/msmc-tools/blob/master/multihetsep_bootstrap.py), after which MSMC2 and MSMC-IM were run on these replicates in the same manner as for the whole genomes (see below).

To scale historic events for the woodland strawberry, the experimentally tested mutation rate of *A. thaliana* of $7.1 \times 10^{-9}$ per bp per generation[53] and a 2-year generation time that has been used for studies on other perennial herbs[51,52] were used. We also tested the alternative mutation rate, the evolutionary $5.6 \times 10^{-9}$ per bp per generation estimated across perennial *Fragaria* species, ($2.8 \times 10^{-9}$ per bp per year)[13]. However, since the mutation rate of *Arabidopsis* provided the highest temporal resolution against GI cycles specifically in *Fragaria vesca* (Supplementary Figs. 10–14), we used it to infer the demographic history of the species.

Using the MSMC-IM, the time-dependent (generations ago) symmetric migration rate (m(t)) and migration rate refined coalescence rates (effective population size, $N_E$) were inferred for each population pair (im_N1 and im_N2). im_N1 was selected to represent historical $N_E$ because the analyses with some sample pairs finished successfully without clear errors only when a specific sample was assigned as im_N1. This was especially the case when one of the samples had a small current $N_E$. In addition, in bootstrap replicates, the current im_N2 was also often inflated for unknown reasons. All MSMC-IM output files (designated as *.estimate-files) underwent a thorough review (Supplementary Data 4). Strong glacial periods, such as PGP, often led to failures of runs (8% of all runs) due to extreme bottlenecks in either one or both populations. Those runs were not

accepted for estimating initial divergence ($M < 0.999$, $m < 0.99$), but were accepted for estimating split times (estimated median split time of a population pair, $M(t) < 0.5$) that occurred during the Holocene. If a run failed during the Holocene, split times were not accepted; however, those runs were accepted to estimate initial divergence, which usually initiated much earlier in history. If a failure occurred more than once, leading to fluctuations of N1 and N2 parameters between extreme values, runs were discarded from results. Such runs often occurred when the current effective population sizes ($N_E$) were low in both populations (indicated by low im_N1 and im_N2 values, as observed in the Alta-Kåfjord populations). However, for assessing current $N_E$, both im_N1 and im_N2 parameters were accepted, given that earlier time points did not have an impact (the principle of Markovian property) on the most recent coalescence rates (Supplementary Data 3). As mentioned earlier, hybrid samples with strongly negative $F_{IS}$ values (ES21, ES3, NOR28, NOR30) were excluded from regional pools. Additionally, two Iberian (ES13 and ES18), one Finnish (FIN12) and one Italian (IT3) samples that showed suspiciously steep recent growth (< 650 generations ago) despite still being highly inbred were excluded.

## Heterozygosity
Regional nucleotide diversities were calculated from imputed and phased polymorphic sites using vcftools[89]. Intergenic sites were used to compare nucleotide diversities between regions. Nucleotide heterozygosity for each sample was estimated with ANGSD-software v.0.918-43[101] directly from prefiltered bam-files using the folded frequency spectrum. The following filters were used: uniqueOnly 1, minQ 20, minmapQ 30 -C50.

## Genotyping by sequencing (GBS)
To explore population level genetic variation in woodland strawberry and to include samples from geographic regions not represented in the WGS dataset, we analyzed 330 samples from 23 natural populations from Finland, Italy, northern Norwegian Alta, Kåfjord and Tromso. In addition, 10 samples were collected from Sweden, along with several individual samples from Finland, some of which overlapped with existing whole-genome data (Supplementary Data 1). Genotyping-by-sequencing (GBS) was carried out at the Cornell University according to Elshire et al.[102]. GBS libraries were sequenced (Supplementary Data 1) using the Illumina HiSeq2500 platform (San Diego, CA, USA). For mapping and variant calling, we used the Tassel 3 pipeline (Github (https://github.com/tuomas64/strawberry)). Raw SNP panel consisted of 94236 variants with an average of 29.3% missing data per variant and mean coverage of 9.82. Using vcftools-v.0.1.17, for the final dataset, we retained only biallelic sites, which had less than 10% missing data, and removed excessively heterozygous sites based on HWE assumption ($p < 0.01$). Excessively homozygote sites were not removed due to self-compatibility of the species. Then, sites with a mean depth of less than six or more than 20 were removed as well as indels. Minor allele frequency was set greater than 0.01, which resulted in 12655 SNPs. Average percentage of missing proportion of site was 1.3%. PCA was constructed using SNPrelate software-v.1.40.0[93] using all SNPs.

## Statistics and reproducibility
Correlations between latitude, effective population size, and inbreeding coefficient were assessed using Pearson's correlation coefficients (R version 4.3.2) based on regional median values of independent samples. The number of replicates per region is indicated in the Fig. 2 ($n \geq 4$, depending on the available number of sequenced samples per region; $p$ values from cor.test). Regional medians were used because they best represent typical populations within each region, some of which encompass highly divergent populations across large geographic areas. Correlations between admixture proportions calculated for individual samples and bioclimatic variables of their collection sites were also tested using Pearson's correlation with Bonferroni-corrected $p$ values (p.adjust) using all samples.

To estimate confidence intervals for demographic trajectories (Fig. 3A–F), replicate MSMC-IM curves were summarized by computing empirical percentiles across biological replicates (unique combinations of

samples from the primary dataset) on a common time grid ($n = 80$ time points). For each time point, we calculated the 50th percentile (median) and the 12.5–87.5% and 2.5–97.5% percentile envelopes. Percentiles for migration rate (m) were computed in log space, and for effective population size ($N_e$) in linear scale. Curves were smoothed using PCHIP (Piecewise Cubic Hermite Interpolating Polynomial) interpolation in log–log space without extrapolation, applied uniformly to both median and percentile series. The number of replicates used is indicated in each figure or figure legend.

Differences in effective population size between peripheral populations (including populations with at least five MSMC-IM runs with the tested population as im_N1, Supplementary Data 4) and southern European populations (Croatia, Italy, Germany) during the MIS2 glaciation (29–14 ka) were tested using the Wilcoxon rank-sum test with FDR correction (Benjamini–Hochberg). For postglacial colonization patterns, Shapiro's test was used to assess the normality of split times for each regional sample pair, followed by $t$-tests or Wilcoxon rank-sum tests as appropriate. FDR-adjusted $p$ values were computed using the Benjamini-Hochberg method (p.adjust).

### Reporting summary
Further information on research design is available in the Nature Portfolio Reporting Summary linked to this article.

### Data availability
Genome sequence and GBS data have been deposited to NCBI (PRJNA1018297 and PRJNA1357314, respectively). All numerical data presented in the article have been deposited in the Dryad repository[103].

### Code availability
The code used for data-analysis is available on Github (https://github.com/tuomas64/strawberry) and Zenodo[104]. Code was developed with the assistance of GPT-4 and GPT-5.

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

## Acknowledgements

The research was funded by the Research Council of Finland (grant nr 317306 for T.H., grant nr 315353 for T.T., and grant nr 331426 for J.S.S.), and European Research Council (ERC-2014-StG 638134) and the Spanish Ministries of Science, Innovation and Universities (PID2021-123677OB-I00) for D.P. The research was also funded through the 2019–2020 BiodivERsA joint call for research proposals, under the BiodivClim ERA-Net COFUND programme, and with the funding organization Research Council of Finland (grant nr 344726 for T.H.). The study included material from the "Professor Staudt Collection" maintained in Dresden, Germany at the Hansabred GmbH & Co. KG and from the IFAPA *Fragaria* collection funded by IFAPA Project PR.CRF.CRF202200.002 of the European Agricultural Fund for Rural Development, within the Rural Development Program of Andalusia 2014–2022. We thank Eija Takala and Marjo Kilpinen for laboratory assistance and Katriina Palm for maintaining the plant materials. Computations were performed using CSC-IT Center of Science—high-performance computing cluster.

## Author contributions

T.T. and T.H. designed research, T.T., P.A., and T.H. performed research, J.S. contributed new analytic tools, T.T., J.S.S., J.K., S.L., A.L., and V.A.A. analyzed data, T.T., J.S.S., H.D.K., V.A.A., and T.H. wrote the paper, D.P., P.P.E., and P.A. contributed new data, T.T., H.S.H., J.H.H., D.J.S, K.O., J.F.S, L.J., J.A.S., B.D., J.L., H.V., and T.H. contributed plant materials.

## Competing interests

The authors declare no competing interests.

## Additional information

[1]Department of Agricultural Sciences, Viikki Plant Science Centre, University of Helsinki, Helsinki, Finland. [2]Department of Geosciences and Geography, University of Helsinki, Helsinki, Finland. [3]Department of Biological Sciences, University at Buffalo, Buffalo, NY, USA. [4]Division of Ecology, Evolution and Biodiversity Conservation, Biology Department, KU Leuven, Leuven, Belgium. [5]Institute of Biotechnology, University of Helsinki, Helsinki, Finland. [6]Department of Horticulture, Michigan State University, East Lansing, MI, USA. [7]Genetics and Genome Sciences Program, Michigan State University, East Lansing, MI, USA. [8]Faculty of Agricultural Sciences, Agricultural University of Iceland, Borgarbyggð, Iceland. [9]Faculty of Agricultural Sciences, Agricultural University of Iceland, Reykjavik, Iceland. [10]Department of Plant Genetics, NIAB, Cambridge, UK. [11]Albrecht Daniel Thaer-Institute of Agricultural and Horticultural Sciences, Humboldt-Universität Berlin, Berlin, Germany. [12]Andalusian Institute of Agricultural and Fisheries Research & Training, IFAPA Center Málaga, Málaga, Spain. [13]Department of Arctic and Marine Biology, UiT The Arctic University of Norway, Tromsø, Norway. [14]Department of Plant Protection Biology, Swedish University of Agricultural Sciences, Lomma, Sweden. [15]Department of Pomology, Faculty of Agriculture, University of Zagreb, Zagreb, Croatia. [16]Laboratory of Economic Botany, Institute of Botany, Nature Research Centre, Vilnius, Lithuania. [17]Pharmacy and Pharmacology Center, Institute of Biomedical Sciences, Faculty of Medicine, Vilnius University, Vilnius, Lithuania. [18]Botanical Museum, Finnish Museum of Natural History, University of Helsinki, Helsinki, Finland. [19]School of Biological Sciences, Nanyang Technological University, Singapore, Singapore. [20]Organismal and Evolutionary Biology Research Programme and Viikki Plant Science Centre, Faculty of Biological and Environmental Sciences, University of Helsinki, Helsinki, Finland. [21]Singapore Centre for Environmental Life Sciences Engineering, Nanyang Technological University, Singapore, Singapore. [22]Departamento de Biología Molecular y Bioquímica, Facultad de Ciencias, Instituto de Hortofruticultura Subtropical y Mediterránea "La Mayora" (IHSM), Universidad de Málaga—Consejo Superior de Investigaciones Científicas (UMA-CSIC), Málaga, Spain. ✉e-mail: timo.hytonen@helsinki.fi

