## [Transparent Peer Review file · Communications Biology]

The Late Quaternary climate impact on the genome of the woodland strawberry (*Fragaria vesca*), a perennial herb

Corresponding Author: Professor Timo Hytönen

Version 0:

Reviewer comments:

Reviewer #1

(Remarks to the Author)

This study investigated paleoclimatic history of a perennial plant species (*Fragaria vesca*) using genome data and found genetic division between its western and eastern range. Together with range-wide genetic structure, this study estimated demographic history and divergence time of populations, and thereby, suggest distinct glacial refugia at eastern and western range and recolonization history. Overall, authors carefully analyzed the genetic structure and demographic history from genome data and lead reasonable conclusion in the history following the past climate changes of the wild strawberry species.

However, the novelty of this study seems to be weak. Although this study provides demographic history including the detailed temporal framework of genetic divergence, the major finding and conclusion solely supported the conventional scenario of paleoclimatic history in plant communities, which has been accumulated by numerous phylogeographic studies in the past 30 years. The manuscript is unclear what is the unanswered issues on the paleoclimatic history of plant communities in Europe, and therefore, the general novelty of this study in biology is unclear.

The second weak point is the lack of analyses on conclusive statements. Authors emphasized the influence of environmental difference on the genetic structure (line 497–500). Elucidating environmental effects on the genetic structure and associated genetic background would provide novel insight by such range-wide genetic study, but authors did not focus on this topic. Statistical analyses on environmental difference between genetic groups and/or association of environment and SNPs enables authors to evaluate the hypothesis and would strengthen the novelty of this study.

Minor comments:

Line 472

In Fig. 4A, the divergence time may be 10–1 ka. 2-1,3 ka needs clarification.

Line 497–500

This conclusive statement needs to be examined by statistical analyses using environmental data.

Reviewer #2

(Remarks to the Author)

In this study, Toivainen et al. present an investigation on the perennial woodland strawberry, *Fragaria vesca*, using whole-genome sequencing data from 200 individuals sampled across large parts of the species' natural range in the western part of Europe. The authors performed a suite of complementary statistical analyses to (i) elucidate the phylogenetic relationships among individuals and regional occurrences, (ii) evaluate the genetic clustering of the samples and possible admixture of lineages, and (iii) infer the demographic history that could explain the current distribution of genetic variation in the species. From the findings, the authors infer a split of two main evolutionary lineages across Europe (Eastern/Western), related to putative refugial areas in the southern European peninsulas during glacial periods, and re-colonization northwards that led to the admixture of the two lineages in areas of secondary contact.

General comments

This manuscript reflects a large effort and an extensive dataset on the woodland strawberry, and the analyses in most parts are thoroughly performed, yielding interesting inference on the Quaternary history of this forest herb species. However, I see several substantial shortcomings that I address in the following comments, but also in the annotations throughout the pdf files of the manuscript including Supplementary material. In their sum, these shortcomings require a substantial investment into this article to make it a conclusive, well-supported study.

1) To infer the demographic history of a species in response to glacial oscillations, I consider it important to represent a broad spatial sampling including putative refugial areas. The re-colonization of the studied range is described as the result of two northward-expanding lineages from southern refugia. However, it remains open to what degree a potential eastern refugium (e.g. the Moscow basin) could have contributed to the current genetic make-up of the northwestern European populations; it is a pity that only few, sparse samples from the northeastern species range in Russia and adjacent countries were included in the sampling, which may have political reasons. Likewise, there are important gaps within the current sampling, like the Balkan peninsula (1 Romanian sample and a few individuals from northern Croatia are insufficient to represent this important potential refugium) and entire Sweden and central/northern Finland (relating to trans-Scandinavian migration from the East or South). Hence, some of the demographic inference on where the species retracted to during cold periods and from where it re-expanded, remains rather speculative.

2) Regarding the fluctuations of the effective population size (N_e) and respective historic migration rates (m), I found it interesting how these somewhat align with the glacial cycles. However, the timing of these trajectories, at least to some degree, depends on the chosen mutation rate (here taken from *Arabidopsis thaliana*). It would be interesting, if not important, to test and report on alternative outcomes when using a range of mutation rates in the demographic model.

3) The authors claim that the coincidental alignment of the phylogenetic split into two evolutionary lineages (Eastern/Western) and of the transition between oceanic vs. continental climate (temperature seasonality) may indicate an adaptive response to this environmental gradient. Given the analyses presented, I consider this assumption rather critical and would certainly want to see a test (e.g. correlation of admixture proportion to one of the clusters vs. temperature seasonality) or an analysis searching for genomic regions showing such a differentiation (e.g. genotype–environment association), at best substantiating an association by the respective underlying biological function of the genomic region concerned.

4) A pattern of isolation by distance is inferred but not explicitly demonstrated. Testing for such a correlation on the entire dataset, and also separately on the eastern and western lineages, would be relevant to make such a claim.

5) I consider the estimation of F_{is} as problematic because individuals are grouped over large spatial extent, which violates the HWE assumption of a panmictic population (or deviation from panmixia). Given the WGS dataset, I would urge the authors to consider an estimation of runs of homozygosity (per individual), as was done, e.g., by Laenen et al. (PNAS 2018; cited in the manuscript). This estimator might be more suitable to infer a latitudinal gradient in levels of inbreeding.

Some further issues to pay attention to are listed below:

6) There is no mention about potential hybridization with cultivated strawberries, if this is possible at all given the different ploidy levels. Or to what degree human transfer could have affected the pattern observed (there are numerous samples taken from urban areas or other settlement, implying that strawberry was (un)intentionally brought to these places from (un)known sources.

7) In the beginning of the Results and Discussion section, it is mentioned that 2.36 Mio SNPs were retained after filtering. In turn, the Material & Methods section refers to >2.7 Mio variants (including indels?), and this data was massively filtered to end up with a few tens of thousands of SNPs used for (all?) analyses. Clarification on the datasets used are required.

8) The authors should explain why they present two alternative estimators of genetic diversity, θ and π , and why the outcomes of correlations involving the two estimators do not show consistent results.

9) Sampling coordinates are not fully correct (Data Set S3): in some cases, they point to sites located in alpine terrain (e.g. Swiss and Austrian sample), in the sea or running water (e.g. Norway, Iceland), in various city halls (e.g. Versailles, Erstein, Gatersleben), in a private house (Merzhausen, Germany) or garden (Russia), between train tracks (Oberstaufen; dangerous for sampling!!), or on a glacier (Iceland). This is particularly surprising because all these coordinates that I checked had 6 decimals, hence should be rather precise.

10) Language: In several instances, I question the appropriate terminology, text contains what I see as overstatements (e.g., “groundbreaking, general insights”), and articles are missing (more rarely: should be deleted). I therefore suggest choosing appropriate, consistent terminology, tone down some of the statements, and do a careful language check.

Felix Gugerli

Reviewer #3

(Remarks to the Author)

In this manuscript, Toivainen and colleagues explored the paleohistory and modern genetic structure of the woodland strawberry (*Fragaria vesca*) across its European range using population genomic approaches. This study identifies distinct western and eastern genetic clusters, shaped by glacial refugia and adaptations to seasonal temperature variations. The authors highlight the value of perennial plants in capturing long-term population genomic signatures that reflect historical climatic events. Overall, this manuscript is well-structured and fit the aims and scope of Communications Biology. However, I have several comments that need to be addressed before it can be further considered for publication.

1) Introduction:

L108: The role of population genomics in resolving the gaps left by paleoecological records is well-addressed, but it would be helpful to expand on how the specific methods used in this study (e.g., whole-genome sequencing) facilitate the understanding of past climatic histories compared to traditional approaches.

L134: The sentence on the comparative studies of perennial herbs like *Arabis alpina* and *Arabidopsis lyrata* could use a clearer transition into the importance of the present study on *F. vesca*. The authors mentioned these species but did not sufficiently explain how their results directly relate to the current research.

L147: The authors claimed that *F. vesca* has a broad geographic distribution in Eurasia and as a non-native subspecies in eastern North America, which is NOT supported by POWO (accessed on Feb 10, 2025). According to POWO, the native range of this species is N. America to Guatemala, Macaronesia, Europe to Siberia and Xinjiang. Any justifications?

L150: The description of *F. vesca*'s life history (both sexual and asexual reproduction) would benefit from a more explicit connection to the study's findings. How do these reproductive strategies influence the species' genetic structure in the context of climatic history?

Meanwhile, a hypothesis-driven narrative would make the introduction more engaging, which is not well achieved in its current form.

2) Results and Discussion:

The mention of hybridization during earlier interglacial periods is intriguing, but this claim needs further clarification. Is there any genetic evidence to support this hypothesis, and how do the authors plan to distinguish between hybridization events and gene flow from distinct refugial populations? Additionally, genotype-environment association (GEA) analysis (such as LFMM and RDA) would add more layers of interesting results to this whole story.

The discussion part could be strengthened by linking the findings to broader conservation issues. For example, what are the practical implications of these findings for the conservation of *F. vesca* or other species in northern temperate regions? Are there any recommendations for preserving genetic diversity in the light of global climate change?

3) Conclusions:

The conclusion part should be more explicit in terms of how the results of this study can be applied to predict future population dynamics in response to climate change. A stronger connection to the broader ecological and evolutionary context would help position the findings as an essential contribution to the scientific community and conservation efforts.

4) Materials and Methods:

L538: Why were these accessions sampled? Are they representative of the natural range across Europe? Are there specific reasons in terms of ecological or genetic aspects?

L554: The use of the *F. vesca* 'Hawaii-4' genome v.4.0 as the reference seems outdated. Recent updates may offer more refined genome assemblies, better annotations, and may have impacts on the results here.

L569: Some additional parameters could be considered when filtering the SNP dataset. It is clear that high-depth site would be subject to false positives but a lower bound of depth should also be considered. Keeping SNP sites with a reasonable DP range (e.g., 25% to 75% in depth quartiles) might help. Meanwhile, assuming all the samples are diploids, additional parameters (--min-alleles 2 --max-alleles 2) to keep these variants bi-allelic are necessary.

L 591: The values of CV error suggest that the studied *F. vesca* accessions do not exhibit strong population stratification. Any justifications?

L595: Adopting the GRT model seems arbitrary. As implemented in IQTREE, I recommend the authors include the -mfp argument to detect the best-fit model.

L602: It might be helpful to discuss how the limitations of the sampling strategy (e.g., not having whole population-level samples) would affect the interpretation of the FIS values and whether these biases were considered in the analysis.

L613: The exclusion of the two samples due to highly negative FIS values is well-justified, but it would be beneficial to explain how recent hybridization was detected briefly. Was it based solely on FIS values, or were other genetic indicators of hybridization used (e.g., admixture analysis)?

L621: While MSMC2 is a widely used approach for demographic history inference, considering the complex population structure of plant species like *F. vesca*, why was an alternative method like fastsimcoal2 not considered? Would MSMC2

perform well in handling varying population sizes across different regions? Are there any limitations or assumptions in using this method, particularly when population sizes differ significantly across regions?

L658: The adoption of the mutation rate of *A. thaliana* seems inaccurate. Given that genomic data of *Fragaria* species are releasing rapidly, I suggest the authors to calculate the mutation rate of *F. vesca* directly or at least adopt it from a *Fragaria* relative.

Version 1:

Reviewer comments:

Reviewer #3

(Remarks to the Author)

This revised manuscript represents a substantial improvement over the previous version I had the opportunity to review earlier this year. I am pleased to note that all of my previous comments have been thoughtfully and thoroughly addressed.

I have no major concerns at this stage. However, I would like to point out two minor issues that may warrant attention:

1. Figure 1G (which is mentioned in the rebuttal letter) appears to be missing from the revised submission.
2. The exact software versions for several tools (e.g., SnpEff, SNPRelate, etc.) are not clearly specified in the Methods section. Providing this information would enhance the reproducibility of the study.

Overall, I find the manuscript to be in good shape and have no further comments.

Reviewer #4

(Remarks to the Author)

In the study "The Late Quaternary climate impact on the wild strawberry genome: the story of a perennial herb", Toivainen and colleagues use WGS data to investigate the recent history of European populations of the perennial wild strawberry, *Fragaria vesca* L. The Authors applied phylogenetic, clustering, admixture and demographic approaches to gain insights in the events that have shaped the current distribution of genetic variation in the species.

As noted by reviewer #1 the study is not entirely innovative, however the work is interesting and the authors demonstrate a solid knowledge of the subject. The suggestions provided by the reviewers have contributed to a substantial improvement of the work compared to the previous version.

Nonetheless, on some occasions the answers given to the questions posed by the referees are not entirely convincing. I see some main problems, already reported by other reviewers regarding the sampling strategy and the demographic inference.

Sampling

The authors claim to have a dense sampling (200 WGS). I would consider a sample of 200 genomes dense if it were related to a city, not over such a large territory as the one studied by the authors. Adding GBS data does not change things Consider other terms. Furthermore, it is not clear what the authors mean by the term "high resolution" in relation to their approach.

As noted by other referees, several regions of Europe have been poorly sampled. Large parts of the Balkan Peninsula have not been sampled. Several European states (Poland, Czech Republic, Slovakia, Hungary...) seem to have been excluded from the sampling without any justification, France is underrepresented and from Italy only north-eastern samples seem to be present (the authors mention samples from the Apennines, which however are not visible on any map). This may have led to ignoring a significant part of the current variability of the species, also because it is known that several areas, especially in southern Europe, served as refugia during glaciations. The authors should therefore take these shortcomings into account in their discussion and assess their potential impact on the results and reconstructions.

The reason why the GBS analysis was done is not clear from the text. Better to add something in M&M

Demographic inference

In their SNP (and indel) identification strategy, the authors chose not to apply a minimum filter to exclude low-coverage sites. They therefore applied several soft filters to try to improve the quality of the data used in the various analyses. Although biologically partially sensible, this strategy can be potentially dangerous, especially for low-variability species, when performing demographic analyses. The risk is to include "high-quality" false positives that, in species with few variable sites, could have a disruptive effect on the reconstructed demography, estimated divergence times and N_e . The authors, could at least consider to filter on the basis of Genotype Quality (GQ) and Phred-scale Likelihood (PL) in order to remove the "riskiest" SNPs and improve the credibility of their results. Comparing results with different filtering strategy (including hard filters) is desirable to evaluate the quality of the strategy adopted by the authors.

The choice not to perform the bootstrap analysis does not seem justified because it prevents evaluating the statistical confidence of the result found. The motivation provided by the authors does not seem convincing. Also, the choice to adopt the *Arabidopsis* mutation rate instead of the one estimated within the genus *Fragaria* should also be better discussed.

In some instances, one has the impression of a circular reasoning carried out by the authors where the result is used to motivate a choice that should be independent of it. Better to avoid it.

Demographic analyses assume absence of selection and population structure. However, as reported by the authors, this could not be the case specially (but not limited to) when the two main groups are analyzed ($k=2$). What impact could this

have on the estimated demographic parameters and demographic histories? The authors should discuss this aspect in depth.

Other points

As suggested by the referee#2 the authors calculated a correlation between the proportions of admixture and some environmental variables. The environmental variables used have a very specific timeframe. Since the admixture events could be traced back a long time ago especially in a species with autogamous and asexual reproduction, I was asking what do these correlations refer to? I think the point should be clarified by the authors.

I think the authors should discuss in more detail what impact the possibility of autogamy and asexual reproduction might have had on the different analyses and what biases they might have introduced into the results.

Apply corrections to significance levels when performing multiple (non-independent) tests

Final recommendation

Overall, I believe that the article deserves to be accepted, but only after the authors clarify the above points and correct several minor details, errors and repetitions in the text. For example, the name of the author of a species should be provided at least the first time the species is named....

Authors are encouraged to reread the entire article carefully before resubmitting it.

Version 2:

Reviewer comments:

Reviewer #4

(Remarks to the Author)

The latest version of the article 'The late Quaternary climate impact on the wild strawberry genome: the story of a perennial herb' represents a substantial improvement over the previous version that had been previously revised.

The authors responded positively to the criticisms raised and clarified the ambiguous points that had been identified. Where potential limitations remain, these are now acknowledged by the authors in the text, thereby improving the transparency of the work.

The methodological choices applied are presented more convincingly and explained in greater detail. I would like to express my gratitude to the authors for the corrections made and congratulate them on their work.

At this stage, there are no significant issues with the work. One small detail needs to be corrected: the authors' names for the species *Malus sieversii* and *Vitis vinifera* ssp. *sylvestris* should now be included in the Introduction.

Reviewer #1 (Remarks to the Author):

This study investigated paleoclimatic history of a perennial plant species (*Fragaria vesca*) using genome data and found genetic division between its western and eastern range. Together with range-wide genetic structure, this study estimated demographic history and divergence time of populations, and thereby, suggest distinct glacial refugia at eastern and western range and recolonization history. Overall, authors carefully analyzed the genetic structure and demographic history from genome data and lead reasonable conclusion in the history following the past climate changes of the wild strawberry species.

However, the novelty of this study seems to be weak. Although this study provides demographic history including the detailed temporal framework of genetic divergence, the major finding and conclusion solely supported the conventional scenario of paleoclimatic history in plant communities, which has been accumulated by numerous phylogeographic studies in the past 30 years. The manuscript is unclear what is the unanswered issues on the paleoclimatic history of plant communities in Europe, and therefore, the general novelty of this study in biology is unclear.

Response:

We acknowledge that the original presentation of our study did not sufficiently highlight its novelty. We have now improved the manuscript in multiple ways and emphasized the novel aspects of our work.

First, accumulated scenarios in the past 30 years have usually not been based on whole genome data. It is therefore important to reconstruct and validate these scenarios across different species using whole-genome data. Whole genome data provides greater resolution for studying paleoclimatic histories.

We have emphasized this point in the Introduction (L116-119):

“With the advent of whole-genome sequencing, the climatic histories of species can now be reconstructed at much higher resolution, offering novel insights into species’

demographic trajectories. In this study, we applied this approach to infer the historical population dynamics in woodland strawberry.”

Second, species with higher N_E have been hypothesized to have higher resilience against glacial periods (Milesi et al. 2024). In this study we test this hypothesis within a perennial herb species and find that eastern European strawberry populations with higher N_e show greater resilience during past glacial periods. This suggests the presence of a large ancestral population in southeastern Europe, potentially in the Balkans. Moreover, these populations do not show evidence for bottlenecks after Holocene Thermal Maximum (6-4 ka ago), when the main divergence of populations occurred across Europe (Fig. 4A) due to habitat fragmentation. In contrast, peripheral populations, which were prone to isolation and bottlenecks during earlier glacial periods, underwent additional bottlenecks likely associated with founder effects following this divergence. Such stark contrasts in population dynamics associated with climatic history between eastern core and western/northern peripheral populations have not been previously demonstrated. These results also suggest that population genetics can be a powerful tool for addressing ecological questions deep into the past.

Third, our results also suggest an unconventional paleoclimatic history (lines 490-506; 528-531), providing support for the controversial cryptic refugia hypothesis. We found evidence for distinct northern microrefugia both in western (NW-refugia, Fig. 1C) and eastern Europe based on earlier split times after the LGM, significantly reduced effective population size during the LGM (Supplementary Data 5), high genetic differentiation (e.g. Alta and Kåfjord, Table S1, Fig. S3B) and a complete cease of migration during the last glacial cycle (Fig. S19C-D) and earlier glacial periods (Fig. 3D, S14D-F, S15C-D). While previous studies have inferred microrefugia using conventional population genetic patterns, such as the absence of isolation by distance, or increased genetic differentiation, our results provide additional temporal resolution to support their existence.

We added this finding to the Conclusions (L572-575):

“Furthermore, small highly differentiated populations in Alta and Kåfjord, that may harbor unique adaptations to the Arctic region, are in a risk of extinction because of rapid warming of the Arctic (Rantanen et al. 2022).”

And also (L578-579):

“Furthermore, cryptic northern refugia likely contributed to the colonization of northern Europe”.

Fourth, the temporal resolution in this study is higher than in previous studies (Sun et al. 2020, Qiao et al. 2021, Dong et al. 2023). Importantly, previous studies have not detected interglacial periods at all. Using MSMC-IM we uncovered a series of events (Fig. 3, S12-17) which aligned with corresponding GI-cycles. This enhanced temporal resolution and sensitivity played a crucial role in drawing conclusions about peripheral versus stable core population patterns. It can also offer a valuable complement to fossil-based studies, particularly for species that are sensitive to climatic perturbations and for which fossil evidence is scarce. However, unlike fossil-based approaches, genetic data alone cannot precisely determine the geographic locations of ancestral populations.

Fifth, previous studies have used reduction in effective population size as a measure to detect past glacial periods. In this study, we demonstrate that the symmetric migration rate (m) provides substantially greater power than effective population size (N_E) for detecting historical climatic events. While N_E -based analyses typically identified at most two glacial periods, m captured signals up to four most recent glacial periods in a single run, at least in this species (Fig. S16, S17).

The fourth and fifth points have been incorporated into the Conclusions (L595-602):

“Finally, our demographic inference using haplotype data in MSMC-IM provided a new level of temporal resolution on the climatic history of perennial herbs. We demonstrated fluctuating N_E and migration rate patterns across four glacial and four interglacial periods over the last 370 ka, with the decreases in migration rate providing higher resolution in detecting glacial maxima compared to the traditionally used N_E .”

Sixth, we present a detailed reconstruction of post-glacial colonization history from both western and eastern refugia, with a level of resolution and statistical support that, to our knowledge, has not been previously demonstrated in any plant species. Our analysis revealed that distinct colonization patterns preserved the east-west clustering of samples across Europe, now shown for the first time with robust statistical evidence.

We emphasize this robust finding in the Conclusions (lines 576–578):

“Robust evidence for multiple refugia was obtained from divergence analyses, which statistically supported colonization of northern Europe from both sides of the continent in multiple waves during the Holocene.”

Lastly, in addition to the new insights into species' paleoclimatic history, our results show that populations are currently connected across a broad geographic range, from northern Spain to southern Finland. This connectivity is important for maintaining genetic variation needed for future adaptation. To our knowledge, no previous studies have reported such extensive population connectivity in a perennial herbaceous species.

We underscore this in Conclusions (L584-586):

“This pattern of population dynamics has not been observed in other perennial herbs and may be linked to efficient seed dispersal in this species.”

Comment:

The second weak point is the lack of analyses on conclusive statements. Authors emphasized the influence of environmental difference on the genetic structure (line 497–500). Elucidating environmental effects on the genetic structure and associated genetic background would provide novel insight by such range-wide genetic study, but authors did not focus on this topic. Statistical analyses on environmental difference between genetic groups and/or association of environment and SNPs enables authors to evaluate the hypothesis and would strengthen the novelty of this study.

Response:

This was an important point to raise and was also highlighted by other reviewers. As one reviewer suggested we calculated correlations of the ADMIXTURE proportion, and all 19 bioclimatic variables based on sample coordinates (Fig. 1F, S4). We found the strongest association of eastern admixture proportions with temperature seasonality associated variables (BIO4 and BIO3). We drafted a new paragraph based on these results (lines 250-257).

Minor comments:

Line 472

In Fig. 4A, the divergence time may be 10–1 ka. 2-1,3 ka needs clarification.

Response:

For clarity, we present the median divergence times between regions (Fig. 4A) rather than the full range of split times. Primary data is found in Supplementary Data 4.

Line 497–500

“Eastern and western glacial refugia and different climatic conditions, such as differences in temperature seasonality, may have had a role in defining this demarcation in woodland strawberry and other species during the Quaternary.”

This conclusive statement needs to be examined by statistical analyses using environmental data.

Response:

After conducting new analyses, which support our earlier conclusion, we have reformulated the conclusive statement to ensure it remains cautious and well-supported by the data (lines 562–564):

“Climatic factors - particularly differences in temperature seasonality - may have contributed to the development of this division.”

Reviewer #2 (Remarks to the Author):

In this study, Toivainen et al. present an investigation on the perennial woodland strawberry, *Fragaria vesca*, using whole-genome sequencing data from 200 individuals sampled across large parts of the species' natural range in the western part of Europe. The authors performed a suite of complementary statistical analyses to (i) elucidate the phylogenetic relationships among individuals and regional occurrences, (ii) evaluate the genetic clustering of the samples and possible admixture of lineages, and (iii) infer the demographic history that could explain the current distribution of genetic variation in the species. From the findings, the authors infer a split of two main evolutionary lineages across Europe (Eastern/Western), related to putative refugial areas in the southern European peninsulas during glacial periods, and re-colonization northwards that led to the admixture of the two lineages in areas of secondary contact.

General comments

This manuscript reflects a large effort and an extensive dataset on the woodland strawberry, and the analyses in most parts are thoroughly performed, yielding interesting inference on the Quaternary history of this forest herb species. However, I see several substantial shortcomings that I address in the following comments, but also in the annotations throughout the pdf files of the manuscript including Supplementary material. In their sum, these shortcomings require a substantial investment into this article to make it a conclusive, well-supported study.

1) To infer the demographic history of a species in response to glacial oscillations, I consider it important to represent a broad spatial sampling including putative refugial areas. The re-colonization of the studied range is described as the result of two northward-expanding lineages from southern refugia. However, it remains open to what degree a potential eastern refugium (e.g. the Moscow basin) could have contributed to the current genetic make-up of the northwestern European populations; it is a pity that only few, sparse samples from the northeastern species range in Russia and adjacent countries were included in the sampling, which may have political reasons. Likewise, there are important gaps within the current sampling, like the

Balkan peninsula (1 Romanian sample and a few individuals from northern Croatia are insufficient to represent this important potential refugium) and entire Sweden and central/northern Finland (relating to trans-Scandinavian migration from the East or South). Hence, some of the demographic inference on where the species retracted to during cold periods and from where it re-expanded, remains rather speculative.

Response:

We acknowledge the limited sampling in Eastern Europe. This is not due to political reasons, but rather a lack of available collaborators in the region. Below, we provide detailed responses to the specific critiques regarding our sampling design.

Potential eastern refugium

To explore the potential role of an eastern refugium in the colonization of northern Europe, we calculated split times from the NW-Russia samples to Alta and Kåfjord (Fig. 4A, Supplementary Data 4). The split times of Alta and Kåfjord samples from northwestern Russian samples were not more recent than from Lithuania (Fig. 4A), suggesting that a large Lithuanian source population (Fig. 3A) was likely a major contributor to Arctic colonization. Russian ancestry likely contributed to the colonization of northeastern Norway (Alta and Porsanger), based on admixture analyses (Fig. S1G). Moreover, our data strongly support the microrefugia hypothesis for these populations, as reviewer 2 suggested. We have incorporated this hypothesis into the population structure section (lines 209–213) and discuss it in detail in the colonization chapter (lines 490–506).

Trans-Scandinavian migration from the East or South

We have added genotyping by sequencing (GBS) data from 343 samples to strengthen the reconstruction of the postglacial colonization history in this species. Population structure unequivocally shows that Swedish samples cluster with Finnish samples and support their southern origin (Figure S2; explained in lines 190-191). We have earlier shown that samples from central and northern Finland cluster with those from southern Finland further supporting a southern origin (Koskela et al. 2017; mentioned in lines 203-205). These findings indicate that Finland, Sweden, and southeastern Norway (Fig. 1B, Fig. S11) were all colonized from a large source

population in Lithuania. This suggests that the Lithuanian source population has repeatedly sent migrants to multiple surrounding satellite regions.

The Balkans refugium

Although our sampling from Croatia and Romania is limited, comprising only five samples and a single individual, respectively, these populations display the highest current effective population sizes (Fig. S6A, S7) and the lowest inbreeding coefficients (Fig. S6B) across all 200 sequenced samples. This, along with their demographic history (Fig. 3, S12-S13), strongly suggests that they, together with Lithuanian samples, represent key ultimate source populations for this species. Notably, these populations appear to have diverged from each other around the same time, approximately 6,000–5,000 years ago (Fig. 4A, S21C-D), when habitats became fragmented. The narrow variance in split times between these panmictic core populations implies that increasing sample sizes in these regions is unlikely to alter our conclusions. Our results support the hypothesis that core populations dispersed northward through eastern Europe (represented by Lithuanian samples) and to a lesser extent through Middle-Europe (Germany), which was colonized mainly from Italy.

Moreover, our approach for estimating split times between source and peripheral regions during the colonization of the north does not require a large number of samples from source regions when some populations in the source region have high effective population sizes. In demographic inference, source populations are typically large, as they maintain high genetic diversity and serve as the origin of migrants during colonization events. Using this reasoning, in western Europe, we selected only the currently largest populations as source populations for northern colonization. For example, even if we had 23 samples from Italy and 21 samples from Iberia, we only used a couple of largest populations (Supplementary Data 4) to estimate of split times related to northern expansion. Our rationale was that these populations most accurately represent ancestral sources of colonization. Including current small populations, whether from Italy or other regions, would likely offer limited value for these estimates, as they function as demographic sinks rather than sources. Moreover, MSMC-IM analysis performs most reliably when at least one of the compared populations had a high current effective population size. The strong

correlation between inbreeding coefficients and population split times (Fig. S11) suggests that colonization occurred in multiple waves. Early colonizers now show the highest levels of inbreeding, whereas more recently established populations show lower inbreeding, likely due to hybridization with earlier-established populations. These correlations support the idea of ongoing gene flow between source populations with high N_E (large Lithuanian, Italian and Spanish populations) and peripheral regions, supporting the validity of our focused approach.

To further support the existence of a stable southeastern source population for this species, it is notable that the largest populations in other plant species are also often found in southeastern Europe. We have now compared our results with other studies and added an alternative hypothesis for our findings (L365-374):

“Although our sampling from southeastern Europe and the Balkans was limited, our results (Fig. S6) align well with earlier studies on wild grapevine (*Vitis sylvestris*, Dong et al. 2023), *Arabidopsis alpina* (Laenen et al. 2018) and primrose (*Primula vulgaris*, Mora-Carrera et al. 2024), all of which report the largest population sizes in southeastern Europe, likely reflecting the region’s role as an important source area for postglacial colonization. An alternative hypothesis that requires further studies is that woodland strawberry has maintained a large stable population in southwestern Asia, for example in the Caucasus region, which may have served as a frequent source of migrants into Europe (Stubbs et al. 2023).”

Although the Balkan Peninsula has been proposed as a potential glacial refugium, current evidence remains inconclusive. Therefore, we have chosen not to include the Balkan refugium in our conclusions to avoid overinterpretation (lines 565-567).

“Peripheral populations were susceptible to isolation and bottlenecks, while current core populations are part of the stable source that persisted throughout multiple Pleistocene glacial cycles”.

It is important to note, however, that all studies on species’ demographic history based on genomic data are inherently somewhat speculative because it is ultimately

impossible to determine historical locations of species and populations even if large sample sizes.

Comment:

2) Regarding the fluctuations of the effective population size (N_e) and respective historic migration rates (m), I found it interesting how these somewhat align with the glacial cycles. However, the timing of these trajectories, at least to some degree, depends on the chosen mutation rate (here taken from *Arabidopsis thaliana*). It would be interesting, if not important, to test and report on alternative outcomes when using a range of mutation rates in the demographic model.

Response:

This suggestion helped us to make this study more reliable. As suggested, we have inferred demographic histories (Fig. S22-23) using three different mutation rates: $5.6e-9$ per nucleotide per generation (calculated earlier for the *Fragaria* genus, Qiao et al. 2021), $7.1e-9$ (*A. thaliana*'s mutation rate, used in this study) and an arbitrary rate of $9.1e-9$. All three rates strongly support our conclusions, with only minor differences affecting the timing and sensitivity of the inferred histories. The mutation rate of *Arabidopsis thaliana* showed the highest resolution and sensitivity (Fig. S22-23) for detecting demographic signals from the last glacial maximum (LGM) and Penultimate Glacial Period (PGP), and for that reason we used it in our analyses. It is important to note that determining the exact mutation rate in woodland strawberry is not possible without extensive long-term experiments (several sexual generations). In addition, generation time and filtering of genomic data also influence on the timing of events. In *Arabidopsis thaliana* the calculated mutation rate (before Ossowski et al. 2010) and the experimentally determined (after Ossowski et al. 2010) deviated from each other.

We greatly appreciated this suggestion, because it also helped us to exclude non-reliable data. Based on different mutation rates, we excluded the most distant data points from the core pattern because they minimally responded to varying mutation rates.

Comment:

3) The authors claim that the coincidental alignment of the phylogenetic split into two evolutionary lineages (Eastern/Western) and of the transition between oceanic vs. continental climate (temperature seasonality) may indicate an adaptive response to this environmental gradient. Given the analyses presented, I consider this assumption rather critical and would certainly want to see a test (e.g. correlation of admixture proportion to one of the clusters vs. temperature seasonality) or an analysis searching for genomic regions showing such a differentiation (e.g. genotype–environment association), at best substantiating an association by the respective underlying biological function of the genomic region concerned.

Response:

We found this suggestion very helpful and, as advised, calculated correlations between the eastern admixture proportion and all 19 bioclimatic variables across samples (Fig. 1F, S4). Temperature seasonality (BIO4) showed the strongest association, the correlation is strongly positive and highly significant. We drafted a new paragraph based on these results (L250-257). As discussed, eastern populations may be adapted to colder climates (Wang et al. 2025). However, functional studies are required to validate this hypothesis.

Comment:

4) A pattern of isolation by distance is inferred but not explicitly demonstrated. Testing for such a correlation on the entire dataset, and also separately on the eastern and western lineages, would be relevant to make such a claim.

Response:

This was an important suggestion and improved our work significantly, providing also support for the northern microrefugia. We have now included IBD using F_{ST} as a measure for genetic distance between regions (Fig. 1E, S3B). IBD is calculated separately for western, eastern and west-east region pairs (105 pairs in total) based on F_{ST} values in Table S1. We found highly significant IBD across all calculations, when the outlier regions, Alta and Kåfjord were excluded from the analysis. Finding Alta and Kåfjord populations as outliers also supports the hypothesis that they originate from separate microrefugia (relicts). Their genetic differentiation from other

regions is much higher than would be expected based on solely post-glacial colonization history, further supporting the microrefugia hypothesis.

From PDF:

Lines 180-183

"This strong geographic patterning likely reflects isolation by distance (IBD) phenomena (Wright 1943, as exemplified by European humans; Novembre et al. 2008), wherein stepwise fixation of SNPs via random genetic drift emerged in postglacially migrating populations."

Comment:

Commonly, such a spatial pattern like is linked not only to random genetic drift, but also to spatially restricted gene flow (with panmixis, no IBD could be detected).

Response:

This was indeed important notion, and we added "restricted gene flow" at the end of the sentence.

"This strong geographic patterning reflects isolation by distance (IBD) phenomena (Wright 1943, as exemplified by European humans; Novembre et al. 2008), wherein stepwise fixation of SNPs via random genetic drift emerged in postglacially migrating populations **with restricted gene flow** (Fig. 1E, S3B)."

Comment:

5) I consider the estimation of F_{is} as problematic because individuals are grouped over large spatial extent, which violates the HWE assumption of a panmictic population (or deviation from panmixia). Given the WGS dataset, I would urge the authors to consider an estimation of runs of homozygosity (per individual), as was done, e.g., by Laenen et al. (PNAS 2018; cited in the manuscript). This estimator might be more suitable to infer a latitudinal gradient in levels of inbreeding.

Response:

This suggestion helped us to make inbreeding coefficients more comparable. As advised, we recalculated inbreeding coefficients based on runs of homozygosity (ROH) for all samples (Fig. 2C-D, S6B, S9A-B), the ROHs were determined with similar approach as in Laenen et al. (2018).

Some further issues to pay attention to are listed below:

Comment:

6) There is no mention about potential hybridization with cultivated strawberries, if this is possible at all given the different ploidy levels. Or to what degree human transfer could have affected the pattern observed (there are numerous samples taken from urban areas or other settlement, implying that strawberry was (un)intentionally brought to these places from (un)known sources.

Response:

Due to the large difference in ploidy levels, hybridization between diploid wild strawberry (*Fragaria vesca*, $2n = 14$) and octoploid cultivated strawberry (*Fragaria × ananassa*, $2n = 56$) is highly unlikely in nature. Experimental crosses between these species typically result in sterile or non-viable offspring due to chromosomal incompatibilities.

Human-mediated transfer can influence genetic patterns, as seen in other plant species, but such cases appear to be rare. Samples collected from the same geographic regions generally cluster genetically with other local samples. However, a few exceptions exist, such as NOR131, which is genetically similar to British samples (Fig. 1C) and stands out as the only western genotype found in southeastern Norway.

Comment:

7) In the beginning of the Results and Discussion section, it is mentioned that 2.36 Mio SNPs were retained after filtering. In turn, the Material & Methods section refers to >2.7 Mio variants (including indels?), and this data was massively filtered to end up with a few tens of thousands of SNPs used for (all?) analyses. Clarification on the datasets used are required.

Response:

We have modified the beginning of Results and Discussion and Materials and methods to show the exact number of variants including SNPs and indels (L176-178):

“We identified 2.72 million biallelic variants (2365994 SNPs + 354732 indels), which were used for population genomic analyses (Supplementary Data 1).”

Massively filtered datasets (few tens of thousands of SNPs, which focused on gene regions at 4-fold degenerate sites and nonsynonymous sites, or linkage disequilibrium filtered SNP set) were used exclusively for population structure analyses, including PCA (Fig. 1B) and maximum-likelihood trees (Fig. 1C). Whole genomes were used to infer the species demographic history (including current effective population size), inbreeding coefficients (both F_{ROH} and F_{IS}), genetic differentiation between regions (F_{st}) and intergenic nucleotide diversity (estimated from approximately half of the genome).

Comment from PDF: either be specific and mention the type of platform, otherwise be more general in referring to next-generation sequencing, not mentioning the brand

Response:

We removed “the Illumina platform.

Comment:

8) The authors should explain why they present two alternative estimators of genetic diversity, θ and π , and why the outcomes of correlations involving the two estimators do not show consistent results.

Response:

This request prompted us to make this section more concise. We removed the sample-specific heterozygosity (θ) estimate from the main results, as it did not provide substantial additional insight into postglacial colonization history beyond other reported statistics. Instead, we added an explanation of nucleotide diversity (π), which

reflects both historical and current effective population sizes (lines 285–288). We also clarified why, for example, NE-Iberian samples exhibit low nucleotide diversity (Fig. S8A) despite currently having relatively high effective population sizes (Fig. 2A).

Comment:

9) Sampling coordinates are not fully correct (Data Set S3): in some cases, they point to sites located in alpine terrain (e.g. Swiss and Austrian sample), in the sea or running water (e.g. Norway, Iceland), in various city halls (e.g. Versailles, Erstein, Gatersleben), in a private house (Merzhausen, Germany) or garden (Russia), between train tracks (Oberstaufen; dangerous for sampling!!), or on a glacier (Iceland). This is particularly surprising because all these coordinates that I checked had 6 decimals, hence should be rather precise.

Response:

Majority of sample coordinates are accurate. However, for some samples received from colleagues, coordinate information was in degree/minute/(second) format. To make the data comparable, we converted these coordinates to decimal format. This conversion often returned several decimals, although the original coordinates were less accurate. Some sample coordinates (Supplementary Data 3) were not accurate enough and were not used when calculating correlations between the admixture proportion and bioclimatic variables.

Comment:

10) Language: In several instances, I question the appropriate terminology, text contains what I see as overstatements (e.g., “groundbreaking, general insights”), and articles are missing (more rarely: should be deleted). I therefore suggest choosing appropriate, consistent terminology, tone down some of the statements, and do a careful language check.

Response:

We have softened the tone as recommended, and the language has been checked and improved throughout the article.

Additional comments on PDFs:

Abstract

“Here, we applied population genomic approaches to understand the paleohistory of a model perennial plant species, woodland strawberry (*Fragaria vesca* L.), throughout its current European range. “

Comment:

Commonly, a model species refers to one that has been extensively studied in various aspects of its biology; here, the authors should rather refer to "study species", which to me seems more appropriate terminology.

Please change throughout the article

Response:

We have changed the “model” species terminology throughout the article.

Comment:

In this study, no specific test for adaptive divergence has been included, which is why I strongly advise the authors to remove, at least temper this statement:

“We surveyed modern genetic structure and demographic history, finding a clear division into western and eastern genetic clusters indicative of distinct glacial refugia and adaptations to cyclical variation in seasonal temperatures”.

Response:

Based on new analyses, we are more confident but still cautious to say:

“East-West admixture proportions of accessions strongly correlated with temperature seasonality of their habitats indicating adaptations to distinct climates in western and eastern Europe.”

“We argue that our forward-looking, fine-scale documentation of woodland strawberry paleogenomic history reflects generally on the future survival of North temperate flora.”

Comment:

I believe this phrasing is a bit misleading, given that there is no modelling/simulating of future (population genetic) processes; please re-phrase.

This terminology implies that ancient DNA samples were analyzed, which is not the case. Hence, re-phrasing seems justified.

Response:

We changed the sentence:

“We argue that our high-resolution reconstruction of woodland strawberry’s climatic history offers insights into the future survival of temperate flora, with broad implications for ecological resilience, climate adaptation, and the temporal dynamics of genome evolution”.

Introduction

Lines 96-99:

“The Northern Hemisphere underwent marked glacial-interglacial (GI) cycles during the Quaternary period, which lasted over the past 2.58 million years (Lisiecki and Raymo 2005). These cyclical warming/cooling climate changes caused repeated distributional expansions and contractions of various species (Hewitt 2000, Melles et al. 2012, Schiferl et al. 2023).”

Comment:

the same applies to the southern hemisphere, so I suggest being a bit broader at least here at the start of the Introduction

Response:

The “Northern Hemisphere” has been replaced with “Earth” and “within northern biotas” removed from the sentence.

Line 100

Comment:

This reference is missing (Schiferl et al. 2023).

Response:

The reference added to the reference list.

Lines 116-118

“Perennial plants, which unlike annuals must live through extreme temperatures year-to-year during glacial periods, hold promise to record long term population genomic signatures relevant across their extended generation times. “

Comment:

This may hold to some degree, but I would argue it's also true for short-lived, even annual species that need to cope with such (short-/long-term) fluctuations.

Response:

We changed as follows: “Perennial plants, which unlike annuals must live through extreme temperatures **year-round**, hold promise to record long term population genomic signatures relevant across their extended generation times”

Lines 125-127

”As such, comparing systems with sensitivity versus resilience to GIs can be expected to usefully illuminate phylogenetic distinctions among perennials within Northern Hemisphere floras.”

Comment:

Phylogenetic appropriate terminology? maybe re-consider

Response:

Phylogenetic replaced with “evolutionary and ecological”

Lines 140-142

F. vesca ssp. vesca L.

Comment:

ssp not in italics

Response:

ssp corrected as normal text

Lines 142-144

”In this study, we adopt the perennial herb woodland strawberry (*F. vesca ssp. vesca* L.) as a model and leverage whole genome sequencing data to investigate the paleo- and modern population structure and GI climatic history from dense sampling across its European range.”

Comment:

As commented above: paleo population structure would rather require analyzing ancient samples (e.g. macro-fossils), hence I would suggest being careful with this terminology.

Response:

We have modified the sentence to better reflect and incorporate our findings (L147-151):

“In this study, we adopt the perennial herb woodland strawberry (*F. vesca ssp. vesca* L.) as a model and leverage whole-genome sequencing data to investigate the population structure through time and its association with GI climatic history from

dense sampling of 200 accessions from across most of the European range of the species.”

Lines 153-158

“Birds and mammals eat the fruit and mediate seed dispersal (Herrera 1989, Heleno et al. 2010, López-Bao and González-Varo 2011). Here, high-resolution genomic analyses uncover a paleoclimatic history of woodland strawberry across its geographical range in Europe and demonstrate colonization patterns that have preserved distinct eastern and western genetic clusters of the species.”

Comment:

There is an abrupt change in topics between these two sentences; please provide a better link to switch from species biology to study questions.

Response:

This was indeed an important improvement to the Introduction, as it allowed us to better integrate species biology with our study questions. We have added several hypotheses about the expected patterns related to our study questions, based on the species' biology:

Lines 156-163:

“Self-fertilization and asexual reproduction are expected to increase genetic structure and inbreeding within the species. In the context of climatic history, predominantly outcrossing (and typically larger) populations are expected to be more resilient against glacial periods although asexual reproduction can provide an advantage in a short-term survival (Holsinger 2000). Sexual reproduction starts in short days in autumn when flower buds are formed.”

Lines 163-168:

“Birds and mammals eat the fruit and disperse seeds (Herrera 1989, Heleno et al. 2010, López-Bao and González-Varo 2011), promoting efficient long-distance migration. This animal mediated seed dispersal likely increases gene flow between populations, contributing greater connectivity of populations within the species' range (De Kort et al. 2021).”

Results and Discussion

Lines 190-192:

"These leading-edge populations also formed distinct groups in the SNP phylogeny and showed high genetic differentiation with an F_{ST} value of 0.77 (Fig. 1C, Table S1), suggesting strong bottlenecks during past colonization."

Comment:

Could such a high level of regional genetic differentiation also result from refugial populations that persisted in the area and were not swamped by re-colonizing, diverged populations?

Response:

This was an excellent suggestion, and we have taken this hypothesis into account by adding (L209-213):

"This refugia hypothesis is further supported by a strongly weakened IBD pattern when these populations are included (Fig. S3B), indicating that their divergence from other regions is greater than would be expected under a simple post-glacial colonization model, as also corroborated by demographic analyses below."

Lines 194-195

"The geographic border between the western (N=107) and the eastern (N=92) European samples,"

Comment:

To me, "border" sounds like a political separation; commonly, we refer to "contact zones" or "suture zones" when referring to areas where diverged genetic lineages admix.

Response:

This was an important clarification, and we have replaced “border” with “contact zone” throughout the article to more accurately reflect regions where diverged genetic lineages admix.

Lines 198-202

”To explore the geographic distribution of western and eastern ancestries in greater detail, we conducted an ADMIXTURE analysis (Alexander et al. 2009). Samples originating from near the border, for example in Croatia, Germany, and central Norway, displayed both eastern and western ancestries (K=2) in their genomes (Fig. 1D, S1G)”

Comment:

It seems that K=2 is the strongest separation, which is often the case in such analyses (cf. Janes et al., Molecular Ecology 2017). Nevertheless, I believe it's reasonable to also describe the situation for higher numbers of K clusters, even more so because the cross-validation curve may be interpreted as giving K=4 as the best-explaining clustering (and K=8 is also shown in the Suppl. Mat.).

Response:

We added a clarification to the sentence (L221-223):

“To explore the geographic distribution of western and eastern ancestries in greater detail, we conducted an ADMIXTURE analysis (Alexander et al. 2009), focusing here on the two major ancestry components (K=2).”

We selected K = 2 for this section to specifically highlight the east–west division of samples, which is central to our current focus. For broader context, we also present results for K = 4, K = 6 and K = 8 in Fig. S1G, while these results are discussed in later sections where they are most relevant.

Lines 210-212

Comment:

“there are several tree species that show a similar genetic divergence, e.g. *Picea abies*, *Abies alba*, ...”

Response:

In this context, we have chosen to highlight only those species that exhibit a highly similar east–west clustering pattern of samples. While *Picea abies* and *Abies alba* do show genetic divergence between their western and eastern lineages, their population structures do not follow the same geographic pattern, from the Adriatic Sea to the Baltic Sea, as observed in our study.

Comment:

It would be simpler to grasp if citations were combined with the respective species to make the direct link.

Response:

We have now combined the citations with the respective species to make the associations clearer and easier to follow.

Line 216

“hybridization zone”

Comment:

Even though several authors use this terminology for intraspecific contact between genetically diverged lineages, I would prefer relating to admixture or contact zone to make the distinction about the taxonomic level.

Response:

As noted earlier, we have replaced the term with 'contact zone' throughout the manuscript to more accurately reflect intraspecific admixture between genetically diverged lineages.

Lines 236-239

"Further, strong negative latitudinal correlations in regional N_E values were found in both western and eastern Europe when the samples from southern and northwestern Iberia were excluded from the analysis (Fig. 2A and 2B)."

Comment:

Here, the authors fail to argue why they exclude parts of the Iberian samples; as such, it seems as they were fishing for correlations, which I consider problematic.

Response:

To avoid confusion, we have added an explanation in the Materials and methods section, clarifying why we focused on a specific region within Iberia (lines 682-687):

"In Iberia, we had the opportunity to focus on a specific geographic region, northeastern Iberia (NE-Iberia) due to enough samples (N=9). Consequently, this region was used as a proxy for the Iberian Peninsula. Other Iberian samples were highly scattered across the peninsula and primarily represented edge-of-range populations."

Lines 248-254

"In contrast, isolated populations in Alta and Kåfjord with very low nucleotide diversity (Fig. S5D) showed negative FIS values (-0.12 and -0.22), suggesting an excess of heterozygous sites within samples, which could be related to asexual reproduction. Excluding Alta and Kåfjord, which were the only regions where FIS did not correlate with NE (Fig. S6), the correlation between NE and FIS was strongly negative and highly significant ($r^2=0.65$, $p < 2.2 \times 10^{-16}$)."

Comment:

because not everyone might be familiar with the northern European geography, mentioning that these areas are located in northern Norway would be helpful.

Response:

We earlier mention that these populations are from northern Norway (L201-202):

”The eastern populations from Kåfjord and Alta, closely located in adjacent fjords in northeastern Norway”

We have removed these speculative interpretations regarding these populations when we replaced regional inbreeding coefficients with ROH-based inbreeding estimates.

Comment:

With their WGS data, the authors could easily check for clonal replicates in their samples. However, given the rather coarse sampling, I would be surprised to see clonally propagated individuals in this set of sequenced individuals. This said, the negative Fis values may rather come from an excess of heterozygosity because samples do not come from the same population.

Response:

It is highly unlikely that our samples include clones, as all accessions were collected from distinct populations. After calculating inbreeding coefficients based on ROHs, we removed all speculation regarding these populations.

Lines 254-256

”Furthermore, a positive latitudinal correlation was found in FIS, especially in the western genetic cluster, ranging from 0.35 in northeastern Iberia to 0.8 in Iceland and Tromsø in northwestern Norway (Fig. 2C and 2D).

Comment:

it is not clear if these values refer to Fis or to the correlation – please specify

Response:

We clarified the sentence (L280) :

"Furthermore, a positive latitudinal correlation was observed in F_{ROH} , especially in the western genetic cluster, F_{ROH} – values ranging from 0.5 in northeastern Iberia to 0.75 in Iceland and Tromsø in northwestern Norway (Fig. 2C and 2D)."

Comments:

1 (lines 280-289): I have difficulties here following the argumentation. First, the sampling only comprises a single Romanian individual, which also displays a rather peculiar make-up (c.f PCA in Fig. 1). So considering this sample as representative of the Balkan population of woodland strawberry seems like sticking one's neck rather far out.

Second, I cannot agree with the grouping of the Romanian sample with the Lithuanian samples to form a group referred to as one of the southern European peninsulas. At least this is what I read from this sentence, so please re-consider and/or specify.

2: As indicated above, I have my doubts about statements referring to THREE southern peninsulas when the Balkan region was hardly sampled. Moreover, I wonder how this statement on three distinct evolutionary lineages aligns with the finding of only two major genetic clusters (c.f. ADMIXTURE analysis)? Here, the authors need to re- consider their interpretation.

Response:

This was a highly valuable critique, and to response to both points, we have reconstructed the whole paragraph (lines 316–332) accordingly. As suggested, the revised paragraph now places greater emphasis on the west–east division of samples at $K=2$. Eastern ancestry in the Romanian sample is supported by the admixture graph (Fig. S1G) and SNP-phylogeny (Fig. 1C).

Comment:

A reasonable attempt, but even more reasonable if the comparison was made with species from the same habitat type (forest ecosystems; e.g. taking results from the many forest tree studies).

Response:

We acknowledge the reviewer's suggestion and agree that comparisons within similar habitat types are ideal. However, it is challenging to find studies from forest ecosystems, particularly plants, that clearly trace divergence back to the MIS 10 glacial period. While brown bear and tawny owl are indeed animals, they inhabit forest ecosystems, and their mitochondrial DNA has been shown to provide highly accurate phylogeographic reconstructions.

Line 384

reaching their "apex"

Comment:

not sure if this term is appropriate in this context; how about "peak" or "highest value"?

Response:

We have replaced it with "maximum".

Lines 408-409

We analyzed for the first time the continent-wide postglacial colonization patterns in an herbaceous perennial plant species

Comment:

While I'm always a bit skeptical about such a "we are the first" statement, I feel like this sentence is out of place here. Consider removing or at least putting it into a context.

Response:

We agree that the statement was out of place and have removed it. We also revised the beginning of the chapter (L445-450) to more clearly describe how we investigated postglacial colonization patterns.

Line 463

individuals from Scotland (UK1 and UK6),

Comment:

At first, I related this abbreviation UK with Ukraine; to avoid such confusion, I would suggest using GB instead for British samples.

Response:

We have used the abbreviation 'UK' consistently from the beginning of the project, as shown at the tips of Fig. 1C (with 'SCO' noted above to indicate Scotland). Additionally, the text explicitly states that these samples are from Scotland, which we hope helps to avoid confusion.

Lines 487-489

should history repeat itself, when populations retreat south during future glacial periods, the Balkan peninsula may yet again serve as a haven for the genetic diversity of the species.

Comment:

This statement sounds very strange to me, in particular because we are currently facing a completely opposite trend (rapid warming and likely poleward movement of species).

Response:

We have revised the statement to better reflect current climate trends, the expected poleward shifts in species distributions, and conservation considerations (L544-554):

“In the face of ongoing climate change and increasing warming at higher latitudes (Rantanen et al. 2022), this population chain is expected to expand northward. However, forest habitat fragmentation, which began approximately 5,000 years ago, now poses a major threat, as most populations are small and vulnerable to extinction. While the current migration network continues to support genetic diversity at a broad scale, it remains uncertain how long this connectivity can persist under ongoing environmental change. Notably, the large Croatian and Romanian populations, though not part of the main migration chain, retain substantial genetic diversity that is likely important for the species’ survival during future GI cycles.”

Line 498:

“and different climatic conditions, such as differences in temperature seasonality”

As indicated above, I would not make a too distinct link between phylogenetic separation (neutral, demographic signal) and coincidental environmental difference/gradient (temperature seasonality) without explicit testing.

Response:

After conducting more analyses (Fig. 1F, S4), we can more confidently but still cautiously conclude:

“Climatic factors - particularly differences in temperature seasonality - may have contributed to the development of this division”

Lines 571-573

“Excessively heterozygous sites were removed from all samples if they deviated from Hardy-Weinberg equilibrium ($p < 0.01$) in any regional set of samples.”

Comments:

As for the estimation of Fis values, I wonder if filtering for excessively heterozygous loci across a regional sample of individuals is well justified.

Response:

We have added more detailed explanation (L651-655), why we removed those sites:

"Excessively heterozygous sites, likely erroneous in this species, where most populations exhibit signs of inbreeding, were removed from all samples if they deviated from Hardy-Weinberg equilibrium ($p < 0.01$) in any regional set of samples. Excessively homozygous sites were not removed because of the self-compatible nature of *F. vesca*."

Lines 591-592

"Cross-validation error was the lowest at $K=8$ (Fig. S1H), which was selected to represent population structure in the admixture data set."

Comment:

Nevertheless, the authors primarily refer to the main split found at $K=2$. And even more so, I would argue to include the trajectory of cross-validation values, i.e., to choose the K value before the curve flattens ($K=4$) as best-explaining option.

Response:

We have now added also $K=4$ and $K=6$ graphs for comparison (Fig. S1G). We also discuss those shortly in appropriate sections of the article.

Lines 613-616

"Two samples from Iberia (ES21 and ES3) and Tromsø (NOR30 and NOR28) were excluded from analyses based on highly negative FIS values compared to other regional values, probably caused by recent hybridization."

Comment:

Which related species would this hybridization involve? Could sample contamination also be a reason for the excess heterozygosity in these samples?

Response:T

We have modified this part:

“Two samples from Iberia (ES21 and ES3) and Tromsø (NOR30 and NOR28) were excluded from analyses based on highly negative F_{IS} values compared to other regional values (Fig. 9C), probably caused by recent admixture (Fig. S1G). In Tromsø, NOR30 and NOR28 showed higher proportion of Kåfjord ancestry compared with other Tromsø samples.”

If hybridization is very recent, genomes can show an excess of heterozygosity. In Tromsø, hybridization has occurred with genetically highly different Kåfjord populations. Thus, two highly different homozygous populations are hybridized, which has resulted in an excess of heterozygous sites. Recombination has not had sufficient time to break down the hybrid genomic structure. In Iberia, the reason is more unclear, why those particular samples show an excess of heterozygosity, it could be caused also by another species.

Line 731:

Comment:

Strix aluco in italics

Response:

We have corrected the formatting and italicized *Strix aluco* as required.

Line 1121:

Magenta color

Comment:

This choice of colors is not optimal for color-blind people (difficult to distinguish between blue and magenta); consider applying an alternative color scheme.

The same applies to Figs. 1 & 2, but it seems difficult working with different symbols given the current distinction of eastern/western lineages.

Response:

Magenta colour in Fig. 4 has been replaced with better colour (orange) for color-blind people.

Supplementary materials:

Text S1:

"For historical N_E and migration rates, we focused on common clear patterns (Fig. S17), that aligned with the chronological timing of serial MIS (Lisiecki and Raymo 2005)"

Comment:

parameters and abbreviations should be explained also in this supplementary text

Response:

We have added explanations of abbreviations also in the supplementary text.

Fig. S3:

Comment:

but why is there a value for the single sample from Karelia? And which sample is this? It's not listed as such in Dataset S3.

Response:

We have recalculated inbreeding coefficients based on runs of homozygosity and these results are now presented in Fig. S6B. That sample is RUS10 from Karelia. To get earlier regional inbreeding coefficient for that individual, it was combined with three northwestern Russian samples.

References:

Dong Y, Duan SC, Xia QJ, Liang ZC, Dong X, Margaryan K, Musayev M, Goryslavets S, Zdunic G, Bert PF, et al. 2023. Dual domestications and origin of traits in grapevine evolution. *Science* 379:892-901.

Koskela EA, Kurokura T, Toivainen T, Sønsteby A, Heide OM, Sargent DJ, Isobe S, Jaakola L, Hilmarsson H, Elomaa P, Hytönen T. 2017. Altered regulation of TERMINAL FLOWER 1 causes the unique vernalisation response in an arctic woodland strawberry accession. *New Phytologist* 216:841-53.

Laenen B, Tedder A, Nowak MD, Toräng P, Wunder J, Wötzel S, Steige KA, Kourmpetis Y, Odong T, Drouzas AD, et al. 2018. Demography and mating system shape the genome-wide impact of purifying selection in *Arabis alpina*. *Proceedings of the National Academy of Sciences of the United States of America* 115:816-821.

Mora-Carrera, E., et al. (2024). Unveiling the Genome-Wide Consequences of Range Expansion and Mating System Transitions in *Primula vulgaris*. *Genome Biology and Evolution* 16(10).

Ossowski S, Schneeberger K, Lucas-Lledó JI, Warthmann N, Clark RM, Shaw RG, Weigel D, Lynch M. 2010. The rate and molecular spectrum of spontaneous mutations in *Arabidopsis thaliana*. *Science* 327:92-94.

Rantanen M, Karpechko AY, Lipponen A, Nordling K, Hyvärinen O, Ruosteenoja K, Vihma T, Laaksonen A. 2022. The Arctic has warmed nearly four times faster than the globe since 1979. *Communications Earth & Environment* 3:168

Stubbs RL, Theodoridis S, Mora-Carrera E, Keller B, Yousefi N, Potente G, Léveillé-Bourret É, Celep F, Kochjarová J, Tedoradze G, Eaton DAR, Conti E. 2023. Whole-genome analyses disentangle reticulate evolution of primroses in a biodiversity hotspot. *New Phytologist* 237:656-71.

Qiao Q, Edger PP, Xue L, Qiong L, Lu J, Zhang YC, Cao Q, Yocca AE, Platts AE, Knapp SJ, et al. 2021. Evolutionary history and pan-genome dynamics of strawberry (*Fragaria* spp.). *Proceedings of the National Academy of Sciences of the United States of America* 118: e2105431118.

Wang S, Li J, Yu P, Guo L, Zhou J, Yang J, Wu W. 2025. Convergent evolution in angiosperms adapted to cold climates. *Plant Communications* 6:101258.

Reviewer #3 (Remarks to the Author):

In this manuscript, Toivainen and colleagues explored the paleohistory and modern genetic structure of the woodland strawberry (*Fragaria vesca*) across its European range using population genomic approaches. This study identifies distinct western and eastern genetic clusters, shaped by glacial refugia and adaptations to seasonal temperature variations. The authors highlight the value of perennial plants in capturing long-term population genomic signatures that reflect historical climatic events. Overall, this manuscript is well-structured and fit the aims and scope of Communications Biology. However, I have several comments that need to be addressed before it can be further considered for publication.

1) Introduction:

Comment:

L108: The role of population genomics in resolving the gaps left by paleoecological records is well-addressed, but it would be helpful to expand on how the specific methods used in this study (e.g., whole-genome sequencing) facilitate the understanding of past climatic histories compared to traditional approaches.

Response:

This suggestion was important for improving the introduction and helped to address the critical question raised by another reviewer regarding how this research differs from previous paleoclimatic studies, which have been accumulated over past the past 30 years. As advised, we emphasized the importance of whole genome sequencing for understanding past climatic histories:

Lines 116-119:

“With the advent of whole-genome sequencing, the climatic histories of species can now be reconstructed at much higher resolution, offering novel insights into species' demographic trajectories. In this study, we applied this approach to infer the historical population dynamics in woodland strawberry.”

Comment:

L134: The sentence on the comparative studies of perennial herbs like *Arabis alpina* and *Arabidopsis lyrata* could use a clearer transition into the importance of the present study on *F. vesca*. The authors mentioned these species but did not sufficiently explain how their results directly relate to the current research.

Response:

We have made a clearer transition to those earlier studies and compared those studies more directly to our study:

Lines 138-143

“Previous studies analyzing alpine rockcress (*Arabis alpina* L.) samples from 17 sites and lyrata rockcress (*Arabidopsis lyrata*) from four populations, revealed strong population structures and stable climatic histories in these species. Their northern populations displayed highly reduced genetic diversity, indicating colonization-associated founder effects (Mattila et al. 2017, Laenen et al. 2018).”

Comment:

L147: The authors claimed that *F. vesca* has a broad geographic distribution in Eurasia and as a non-native subspecies in eastern North America, which is NOT supported by POWO (accessed on Feb 10, 2025). According to POWO, the native range of this species is N. America to Guatemala, Macaronesia, Europe to Siberia and Xinjiang. Any justifications?

Response:

We have corrected the statement to better reflect the accepted native range based on POWO (accessed February 10, 2025). The revised sentence (lines 151–154) now reads:

“This major crop wild-relative of the Rosaceae family thrives in diverse habitats, including forests, meadows, and disturbed areas such as roadsides, and has a broad

geographic distribution in Eurasia and North America (Hilmarsson et al. 2017, POWO 2024).”

Comment:

L150: The description of *F. vesca*'s life history (both sexual and asexual reproduction) would benefit from a more explicit connection to the study's findings. How do these reproductive strategies influence the species' genetic structure in the context of climatic history?

Response:

This suggestion also improved the Introduction, and now different reproductive strategies have been put into the context of species' climatic history (L157-161):

”In the context of climatic history, predominantly outcrossing (and typically larger) populations are expected to be more resilient against glacial periods although asexual reproduction can provide an advantage in a short-term survival (Holsinger 2000).”

Comment:

Meanwhile, a hypothesis-driven narrative would make the introduction more engaging, which is not well achieved in its current form.

Response:

We have added several hypotheses to the Introduction regarding expected population structure (L157-158), climatic history (L158-162), and population connectivity (L164-169) based on the species' biology.

2) Results and Discussion:

Comment:

The mention of hybridization during earlier interglacial periods is intriguing, but this claim needs further clarification. Is there any genetic evidence to support this hypothesis, and how do the authors plan to distinguish between hybridization events and gene flow from distinct refugial populations?

Additionally, genotype-environment association (GEA) analysis (such as LFMM and RDA) would add more layers of interesting results to this whole story.

Response:

This was an important point to raise, because we cannot distinguish gene flow from hybridization in the deep past. Accordingly, we have replaced the term “hybridization” with the more neutral terms “contact” or “recontact” throughout the article.

We have conducted more detailed analysis regarding to genotype-environment associations. We calculated correlations with the eastern admixture proportion and all 19 bioclimatic variables across samples as suggested by another reviewer (Fig. 1G, S4). Temperature seasonality (Bio4) and isothermality (Bio3) showed the strongest associations with the proportion which might be linked to adaptation to colder climates in Eastern Europe (Wang et al. 2025). We constructed a new paragraph for this genotype-environment analysis (lines 250-257).

Comment:

The discussion part could be strengthened by linking the findings to broader conservation issues. For example, what are the practical implications of these findings for the conservation of *F. vesca* or other species in northern temperate regions? Are there any recommendations for preserving genetic diversity in the light of global climate change?

Response:

We agree that linking our findings to broader conservation issues adds valuable context. We have addressed the conservation implications of our results at the end of the Results and Discussion section (lines 548–553), including recommendations for preserving genetic diversity in light of ongoing climate change.

“In the face of ongoing climate change and increasing warming at higher latitudes (Rantanen et al. 2022), this population chain is expected to expand northward. However, forest habitat fragmentation, which began approximately 5,000 years ago, now poses a major threat, as most populations are small and vulnerable to extinction.

While the current migration network continues to support genetic diversity at a broad scale, it remains uncertain how long this connectivity can persist under ongoing environmental change. Notably, the large Croatian and Romanian populations, though not part of the main migration chain, retain substantial genetic diversity that is likely important for the species' survival during future GI cycles.“

3) Conclusions:

The conclusion part should be more explicit in terms of how the results of this study can be applied to predict future population dynamics in response to climate change. A stronger connection to the broader ecological and evolutionary context would help position the findings as an essential contribution to the scientific community and conservation efforts.

Response:

We have strengthened the conclusion by adding a statement that highlights the conservation relevance of our findings (L570-575):

” From a conservation perspective, preserving current core populations and selected chain populations with high N_E is essential for maintaining long term genetic diversity of the species. Furthermore, small highly differentiated populations in Alta and Kåfjord, that may harbor unique adaptations to the Arctic region, are in a risk of extinction because of rapid warming of the Arctic (Rantanen et al. 2022)”.

4) Materials and Methods:

L538: Why were these accessions sampled? Are they representative of the natural range across Europe? Are there specific reasons in terms of ecological or genetic aspects?

Response:

These accessions largely represent the natural range of woodland strawberry across Europe. However, there are some gaps in our sampling because of practical reasons, particularly in eastern Europe. In our revised manuscript, we have added genotyping

by sequencing data on Swedish samples, and we refer to earlier study that included broader sampling from Finland (Koskela et al. 2017).

Comment:

L554: The use of the *F. vesca* 'Hawaii-4' genome v.4.0 as the reference seems outdated. Recent updates may offer more refined genome assemblies, better annotations, and may have impacts on the results here.

Response:

We are confident that a newer reference sequence would not significantly enhance the value of this study. Available reference genome (Edger et al. 2017) is already high quality and sufficient for making robust conclusions about the climatic history and population structure of the species. Reanalysing all the data, starting from read mapping, would be too much work at this stage.

Comment:

L569: Some additional parameters could be considered when filtering the SNP dataset. It is clear that high-depth site would be subject to false positives but a lower bound of depth should also be considered. Keeping SNP sites with a reasonable DP range (e.g., 25% to 75% in depth quartiles) might help. Meanwhile, assuming all the samples are diploids, additional parameters (`--min-alleles 2 --max-alleles 2`) to keep these variants bi-allelic are necessary.

Response:

As described in Materials and methods, we inferred genotypes initially using GATK joint calling function. This sophisticated approach uses a Bayesian model to jointly estimate the most likely genotype, homozygous reference, heterozygous, or homozygous alternate, for each sample at every variant site, based on genotype likelihoods, read depth, allele balance, and quality metrics across all individuals. It does not call genotypes with highly uncertain data, which we imputed for sites with less than 10% of missing data. Since we conducted haplotype based demographic modeling, we aimed to retain as much of the genome as possible. Excluding sites with low coverage would have reduced genome coverage and disrupted haplotype

structure in this species, where the average proportion of heterozygous SNPs is very low ($\theta = 0.00059$ per nucleotide per sample, estimated directly from bam files, Supplementary Data 3). In total, low coverage genotype calls (< depth 4) consisted of 3.0% of all genotype calls in the final haplotype data. Thus, we strongly relied GATK genotype calls and chose an approach that tolerated a few errors in order to preserve maximum extent and the integrity of haplotypes. While site-frequency spectrum (SFS)-based methods would indeed require strict filtering of low-coverage sites, because it is essential to have high quality SFS, our approach relied more on intact haplotypes with maximum extent.

To ensure data quality, we applied multiple layers of softer filtering. In addition to GATK based (1) filtering of variants, we removed variants close to indels (2) and excessively heterozygous sites (3). For MSMC2 analyses, we masked low quality regions on a per sample basis (4), for the genome (5), and excluded SNPs with ambiguous phasing (6), which help control for erroneous SNP calls. In population structure analyses, we excluded sites with a minor allele frequency (MAF) < 0.01 (<5/404), removing rare variants that may have arisen from random sequencing errors. For lower quality genotype by sequencing data, where a minimum coverage depth is more critical to ensure reliability of the data, we applied a minimum depth threshold (line 842). Based on our results regarding current (Fig. 2A-B, S6A) and past effective population sizes (Fig. 3, S12-15), our approach appears to have performed well, even though we did not apply the alternative method.

We have already applied the suggested parameters (`--maf > 0.001`, `--max-alleles 2`) in our analyses, but had not explicitly mentioned them in the Materials and Methods. We have now added this information to that section for clarity. While we did refer indirectly to this filtering in the Results and Discussion by stating, “We identified 2.36 million biallelic single nucleotide polymorphisms (SNPs), which were used for population genomic analyses (Supplementary Data 1),” we agree that the parameters should be clearly specified in the methods.

Comment:

L 591: The values of CV error suggest that the studied *F. vesca* accessions do not exhibit strong population stratification. Any justifications?

Response:

ADMIXTURE assumes discrete population structure, but our dataset includes substantial admixture along the contact zone separating the western and eastern genetic clusters across Europe. This continuous genetic exchange possibly contributes to the gradual decline in cross-validation (CV) error with increasing K, and the absence of a clearly defined optimal K value. To provide a broader comparison, we have now included ADMIXTURE plots for K = 4 and K = 6 (Fig. S1G). Despite the complex admixture patterns, both F_{ST} and PCA results indicate strong underlying population structure.

Comment:

L595: Adopting the GRT model seems arbitrary. As implemented in IQTREE, I recommend the authors include the -mfp argument to detect the best-fit model.

Response:

The best-fit substitution model for each dataset has now been selected using the Bayesian Information Criterion (L680-683).

Comment:

L602: It might be helpful to discuss how the limitations of the sampling strategy (e.g., not having whole population-level samples) would affect the interpretation of the F_{IS} values and whether these biases were considered in the analysis.

Response:

As suggested by another reviewer, we recalculated inbreeding coefficients based on runs of homozygosity (ROH) across all samples (Fig. 2C–D, Fig. S6B). These results are comparable across samples. For F_{IS} inbreeding coefficients, we only present results of Croatian and Lithuanian panmictic regions and discuss those shortly. It would be too risky to draw conclusions about population level inbreeding levels based

on region specific F_{IS} values. F_{IS} values are relative for each region and don't reflect sufficiently accurately individual populations within the regions.

Comment:

L613: The exclusion of the two samples due to highly negative F_{IS} values is well-justified, but it would be beneficial to explain how recent hybridization was detected briefly. Was it based solely on F_{IS} values, or were other genetic indicators of hybridization used (e.g., admixture analysis)?

Response:

In those regions, it is also evident from the admixture graphs (Fig. S1G). For example, in Iberia, all other samples consist almost solely of Spanish cluster. However, recent hybridization cannot be detected solely based on admixture graphs, some samples do not show an excess of heterozygosity (i.e. in Croatia or NOR-SE) despite having clearly distinct clusters in their genomes. Also, current effective population sizes in those particular samples are much higher than in other samples (Supplementary Data 1). In most extreme case, in northern Norwegian Tromsø NOR30 sample, current effective population size was over 200 000 suggesting recent hybridization with Kåfjord, which is evident in the admixture graph (Fig. S1G) and the highest sample specific nucleotide heterozygosity across all samples ($\theta=0.002$).

Comment:

L621: While MSMC2 is a widely used approach for demographic history inference, considering the complex population structure of plant species like *F. vesca*, why was an alternative method like fastsimcoal2 not considered? Would MSMC2 perform well in handling varying population sizes across different regions? Are there any limitations or assumptions in using this method, particularly when population sizes differ significantly across regions?

Response:

We selected MSMC2 because our sampling was sparse in some regions: we had only few or a single individual available from some regions. Thus, it would be impossible to detect demographic histories of those samples or regions using the site frequency spectrum-based methods. One important advantage of MSMC2 is, that it does not

require any predefined model, which means that also not predefined (unimaginable) histories can be discovered. Limitation is that we cannot formally test best-fit model - against predefined model. However, multiple sample pairs between regions make results more robust (Fig. S18E) showing a wide range of models (patterns). For example, stable demographic history was significantly more frequent in haplotype combinations between core samples (=best fit model) than non-core haplotype combinations. Also, one caveat is, the method does not work without errors, if both N1 and N2 are currently low.

MSMC2 performed well in capturing variation in current effective population sizes across regions (Fig. 2, S6), as demonstrated by strong latitudinal correlations. As we suggest in the revised manuscript (lines 292–297), the current MSMC-IM estimate of N_E based on the most recent coalescence rate, may more accurately reflect the present state of the population than nucleotide diversity - based estimates, which are known to respond more slowly to demographic changes, as demonstrated in human studies. However, this estimate can be sensitive to hybridization, recently admixed individuals with negative F_{IS} values exhibit inflated effective population sizes (Supplementary Data 3). Additionally, Lithuanian populations showed exceptionally high N_E during the Holocene Thermal Maximum (Fig. 3A), which could also result from admixture with a third population, as MSMC-IM accounts for migration between studied pairs when inferring N_E . Moreover, in the distant past, estimated effective population sizes appeared suspiciously similar across populations, which may reflect methodological bias. Importantly, variation in migration rates provided greater resolution for detecting glacial periods based on isolation events among ancestral lineages than effective population size estimates. This has not been presented earlier and represents important finding of the study.

These criteria to select MSMC2 were added:

Lines 303-304

To explore the demographic history of woodland strawberry, we analyzed population divergence and calculated time-dependent coalescence rates from whole, diploid genomes using the Multiple Sequentially Markovian Coalescent 2 (MSMC2) software (Schiffels and Durbin 2014), which does not require predefined demographic model.

Lines 334-336

We inferred the demographic history of woodland strawberry using haplotype data, an approach that provides high temporal resolution, even with limited sample sizes.

Comment:

L658: The adoption of the mutation rate of *A. thaliana* seems inaccurate. Given that genomic data of *Fragaria* species are releasing rapidly, I suggest the authors to calculate the mutation rate of *F. vesca* directly or at least adopt it from a *Fragaria* relative.

Response:

This was an important suggestion to improve the manuscript. To address this (as also suggested by another reviewer), we tested three different mutation rates (Fig. S22–23), including an estimate from the *Fragaria* genus (5.6×10^{-9} per two years; Qiao et al. 2021). All three rates produced highly comparable demographic results. However, the mutation rate derived from *Arabidopsis thaliana* aligned best with the timing of the two most well-documented glacial periods, the Last Glacial Maximum (22–17 kya) and the Penultimate Glacial Period (190–130 kya), while also maintaining high sensitivity in detecting these events. It is important to note, that it is impossible to know the exact mutation rate of woodland strawberry without extensive long-term experiments (several sexual generations), because also generation time and filtering of the genomic data have influence on the timing of events. For example, in *Arabidopsis thaliana* the inferred mutation rate (before Ossowski et al. 2010) and the experimentally determined (after Ossowski et al. 2010) deviated from each other.

We greatly appreciated this suggestion, because it also helped us to exclude non-reliable data. Based on different mutation rates, we excluded the most distant data points from the core pattern because they minimally responded to varying mutation rates.

References:

Edger PP, VanBuren R, Colle M, Poorten TJ, Wai CM, Niederhuth CE, Alger EI, Ou SJ, Acharya CB, Wang J, et al. 2017. Single-molecule sequencing and optical mapping yields an improved genome of woodland strawberry (*Fragaria vesca*) with chromosome-scale contiguity. *Gigascience* 7:2.

Koskela EA, Kurokura T, Toivainen T, Sønsteby A, Heide OM, Sargent DJ, Isobe S, Jaakola L, Hilmarsson H, Elomaa P, Hytönen T. 2017. Altered regulation of TERMINAL FLOWER 1 causes the unique vernalisation response in an arctic woodland strawberry accession. *New Phytologist* 216:841-53.

Ossowski S, Schneeberger K, Lucas-Lledó JI, Warthmann N, Clark RM, Shaw RG, Weigel D, Lynch M. 2010. The rate and molecular spectrum of spontaneous mutations in *Arabidopsis thaliana*. *Science* 327:92-94.

Wang S., Li J., Yu P., Guo L., Zhou J., Yang J., and Wu W. (2025). Convergent evolution in angiosperms adapted to cold climates. *Plant Comm.* 6, 101258.

Responses to reviews

Reviewer #3 (Remarks to the Author):

This revised manuscript represents a substantial improvement over the previous version I had the opportunity to review earlier this year. I am pleased to note that all of my previous comments have been thoughtfully and thoroughly addressed.

I have no major concerns at this stage. However, I would like to point out two minor issues that may warrant attention:

Comment:

1. Figure 1G (which is mentioned in the rebuttal letter) appears to be missing from the revised submission.

Response:

Figure 1G does not exist, it was a typing error. Fig. 1E and 1F were added to the revision 1 and are also included in this revision. They are correctly cited now.

Comment:

2. The exact software versions for several tools (e.g., SnpEff, SNPRelate, etc.) are not clearly specified in the Methods section. Providing this information would enhance the reproducibility of the study.

Response:

Thank you for pointing this out. The exact software versions have been added for each software used.

Comment:

Overall, I find the manuscript to be in good shape and have no further comments.

Response:

Thank you very much for your constructive feedback.

Reviewer #4 (Remarks to the Author):

In the study “The Late Quaternary climate impact on the wild strawberry genome: the story of a perennial herb”, Toivainen and colleagues use WGS data to investigate the recent history of European populations of the perennial wild strawberry, *Fragaria vesca* L. The Authors applied phylogenetic, clustering, admixture and demographic approaches to gain insights in the events that have shaped the current distribution of genetic variation in the species.

As noted by reviewer #1 the study is not entirely innovative, however the work is interesting and the authors demonstrate a solid knowledge of the subject. The suggestions provided by the reviewers have contributed to a substantial improvement of the work compared to the previous version.

Nonetheless, on some occasions the answers given to the questions posed by the referees are not entirely convincing. I see some main problems, already reported by other reviewers regarding the sampling strategy and the demographic inference.

Sampling

Comment:

The authors claim to have a dense sampling (200 WGS). I would consider a sample of 200 genomes dense if it were related to a city, not over such a large territory as the one studied by the authors. Adding GBS data does not change things. Consider other terms.

Response:

Thank you for pointing this out. We agree and have removed the word 'dense' from the sentence.

Comment:

Furthermore, it is not clear what the authors mean by the term "high resolution" in relation to their approach.

Response:

With "high-resolution," we refer specifically to the high temporal resolution of the reconstructed demographic trajectories that is visible in Figures (Fig. 3, S19). We have clarified this in the abstract (lines 86–89).

"Our reconstruction of woodland strawberry's climatic history with high temporal resolution reveals how late Pleistocene core-periphery dynamics shaped the survival and genome evolution of temperate flora under climate change."

Comment:

As noted by other referees, several regions of Europe have been poorly sampled. Large parts of the Balkan Peninsula have not been sampled. Several European states (Poland, Czech Republic, Slovakia, Hungary...) seem to have been excluded from the sampling without any justification, France is underrepresented and from Italy only north-eastern samples seem to be present (the authors mention samples from the Apennines, which however are not visible on any map). This may have led to ignoring a significant part of the current variability of the species, also because it is known that several areas, especially in southern Europe, served as refugia during glaciations. The authors should therefore take these shortcomings into account in their discussion and assess their potential impact on the results and

reconstructions.

Response:

We agree that the sampling density is not uniform across regions, and that certain areas are indeed underrepresented. We have one sample from Czech Republic (CZ1), which clustered with the eastern cluster (Fig.1C). By 'the Apennines,' we were referring to the Apennine Peninsula. However, our sampling was limited to northern Italy. We modified the sentence to describe this in more detail (lines 196-199):

“The samples from the southern European peninsulas, Iberia (Spain and Portugal), **Apennine Peninsula (northern Italy)**, and the Balkans (Croatia), formed separated groups along the western branch, with the Croatian samples grouping closest to the eastern branch.”

We added a sentence about the uncertainty of the geographic origin of the stable, high-Ne populations in the Discussion (lines 380–388):

“Although broader sampling of **southern European habitats** is needed to draw firm conclusions about the geographic origin of core populations, our results (Fig. S6-7) align well with earlier studies on e.g. alpine rockcress (Laenen et al. 2018), primrose (*Primula vulgaris* **Huds.**, Mora-Carrera et al. 2024), **grey wolf (*Canis lupus*, Mergeay et al. 2024), and dunnock (*Prunella modularis*, Drovetski et al. 2018), all of which show their largest population sizes in the Balkans. This pattern likely reflects the region’s role as a major source area for postglacial colonization, consistent with Hewitt’s refugial paradigm (Hewitt, 1999, Hewitt 2000, Willis et al. 2004, Hewitt 2011).”**

Comment:

The reason why the GBS analysis was done is not clear from the text. Better to add something in M&M.

Response:

We have added a clarification to the beginning of the GBS paragraph in the Materials and Methods section (lines 821–824):

“**To explore population level genetic variation in woodland strawberry and to include samples from geographic regions not represented in the WGS dataset,** we collected 330 samples from 23 natural populations from Finland, Italy, northern Norwegian Alta, Kåfjord and Tromsø. In addition, 10 samples were collected from Sweden, along with several individual samples from Finland, some of which overlapped with existing whole-genome data (Supplementary Data 8).”

Demographic inference

Comment:

In their SNP (and indel) identification strategy, the authors chose not to apply a minimum filter to exclude low-coverage sites. They therefore applied several soft filters to try to improve the quality of the data used in the various analyses. Although biologically partially sensible, this strategy can be potentially dangerous, especially for low-variability species, when performing

demographic analyses. The risk is to include "high-quality" false positives that, in species with few variable sites, could have a disruptive effect on the reconstructed demography, estimated divergence times and N_e . The authors could at least consider to filter on the basis of Genotype Quality (GQ) and Phred-scale Likelihood (PL) in order to remove the "riskiest" SNPs and improve the credibility of their results. Comparing results with different filtering strategy (including hard filters) is desirable to evaluate the quality of the strategy adopted by the authors.

Response:

This was an important point to rise and helpful for reviewing the original filtering protocol. We noticed that we had actually used more stringent sample-specific masks, but this was not correctly described in the manuscript because of our incomplete notes. The script we used (<https://github.com/stschiff/msmc-tools/blob/master/bamCaller.py>) automatically excludes sites with coverage below half of the mean and above twice the mean. To confirm this, I recreated the mask files using the same filtering settings (-q 20 -Q 20) and the original BAM file mean depths and found that the resulting masks matched those generated originally. Thus, a minimum coverage threshold had originally been applied to all samples based on their genome-wide mean coverage. Although this approach, where potentially erroneous SNPs were excluded post hoc, is not optimal, it has improved the robustness of our SNP dataset against potential sequencing errors and miscalled heterozygous sites. All mask files used in this article have been deposited in the Dryad repository (available upon request; becomes publicly available upon acceptance of the article).

In the manuscript, we removed the paragraph in the M&M discussing haplotype intactness and potential errors in heterozygous sites, that was added in the previous version in response to Reviewer 3's request for additional filtering (e.g., minimum depth). Instead, we added a sentence about the importance of stringent filtering and included a detailed description of the bamCaller.py filtering protocol in the Materials and Methods chapter (lines 712–726, see below). We hope this addresses Reviewer 4's concerns regarding the validity of called SNPs for demographic modelling.

Lines 712-726: "To ensure the highest possible quality of the SNP data for coalescence analyses, as recommended by simulation-based evaluations (Sellinger et al. 2021), SNPs were post-filtered with recommended additional filters for each sample (bam-files) and universally for the genome. Specifically, the bamCaller.py script (<https://github.com/stschiff/msmc-tools/blob/master/bamCaller.py>) was used to produce additional sample-specific masks for low quality SNPs potentially produced by the pipeline. To ensure high-confidence site calls and reduce false positives, particularly important in this species with low levels of heterozygosity (Supplementary Data 3), we used the bamCaller.py script with base quality >20 and mapping quality >20 as input parameters. The script's default settings were used to retain only sites with sequencing depths between 0.5× and 2× of the mean depth of each sample's BAM file. Across all samples, the average minimum and maximum depth thresholds were 9.2 and 36.6, respectively, with only eight samples falling below the minimum depth threshold of six (Supplementary Data 1)."

Comment:

The choice not to perform the bootstrap analysis does not seem justified because it prevents evaluating the statistical confidence of the result found. The motivation provided by the authors does not seem convincing.

Response:

In our previous manuscript version, we showed demographic inferences of multiple individual sample pairs in the main text and supplemental figures, which did not allow the estimation of their statistical confidence. In this revision, we provide new figures showing median demographic trajectories with confidence intervals in well-justified sample sets (Fig 3A-F, S10-14). In addition, we provide bootstrap analyses of selected samples for comparison.

Our primary dataset consists of samples with the lowest inbreeding coefficients (Fig. S25–S26) that provide the most robust support for demographic trajectories (Fig. 3A–G). These analyses revealed consistent patterns across diverse genetic backgrounds throughout deep evolutionary history. Additional datasets in supplemental files provide further support for these trajectories (Fig. S10-14). Overall, our results support the presence of several distinct refugia during the prolonged PGP (~60 ka) and MIS 8 (~57 ka) glaciations, in addition to the relatively short MIS 2 glaciation (~15 ka).

For the bootstrap analysis, we followed the original approach (Wang et al. 2020) with higher number of replicates and a chromosome block size scaled with the genome size. We conducted the analysis for eight sample pairs from our primary data (Fig. S25-26), four representing a core pattern (Fig. S17) and the other four a peripheral pattern (Fig. S18), by resampling thirty 1 Mb blocks from seven chromosomes (in total 210Mb) 100 times with the `multihetsep_bootstrap.py` script (https://github.com/stschiff/msmc-tools/blob/master/multihetsep_bootstrap.py). MSMC2 and MSMC-IM analyses were then performed on these pseudochromosomes. The results across all bootstrap replicates supported our original findings (Fig. 3G, S21A-B), showing an excess of isolation events during the PGP and MIS 8 glaciations in the peripheral pattern (Fig. 3H), and during MIS 2 in both patterns (Fig. 3H, S21C).

For the core pattern we added (lines 369-371):

“Bootstrap analyses supported this pattern, indicating that isolation events and bottlenecks were rare between MIS 10 and the LGM (Fig. S17, S21C).”

For the peripheral pattern we added (lines 395-406):

“Consistent with whole-genome results, the bootstrap analysis (400 runs) also revealed increased isolation frequencies during the prolonged PGP (~60 ka) and MIS 8 (~57 ka), and possibly during MIS 10 with less accurate timing due to lower MSMC-IM resolution in the distant past (Fig. 3G-H; Fig. S18). Temporal deviations between bootstrap replicates and whole-genome inferences may result from heterogeneous selection across the genome, including genetic hitchhiking (Smith & Haigh 1974; Gillespie 2000; Schrider et al. 2016) and background selection (Charlesworth et al. 1993; Johri et al. 2021). These effects are expected to be the strongest in genomic regions of low recombination and in predominantly selfing or asexual populations

(Charlesworth et al. 1993), where reduced effective recombination magnifies the impact of linked selection and increases variability in evolutionary rates across the genome.”

In summary, the bootstrap analyses offered increased support for multiple isolation events in the peripheral pattern, despite minor shifts in their timing across replicates.

Comment:

Also, the choice to adopt the *Arabidopsis* mutation rate instead of the one estimated within the genus *Fragaria* should also be better discussed. In some instances, one has the impression of a circular reasoning carried out by the authors where the result is used to motivate a choice that should be independent of it. Better to avoid it.

Response:

We agree that our choice of use of *A. thaliana* mutation rate should be better discussed. To response for the request, we analysed the MSMC2 results (Supplementary Data 4) using also the *Fragaria* mutation rate. We expanded the discussion on our choice to adopt the *Arabidopsis thaliana* mutation rate in the main text (lines 309–319):

“We assumed a two-year generation time, as previously used for other perennial herb species (Savolainen and Kuittinen 2011, Koch et al. 2006) and tested, in multiple datasets, both the experimentally determined mutation rate of *Arabidopsis thaliana* (7.1×10^{-9} mutations per nucleotide per generation; Ossowski et al. 2010; see also <https://www.nature.com/articles/s41588-019-0442-7>) and the evolutionary mutation rate estimated for the *Fragaria* genus. This calibration showed that *Arabidopsis* mutation rate provided a more accurate alignment of demographic events with the timing of past GI cycles (Lisiecki and Raymo 2005) than the evolutionary mutation rate estimated for the whole *Fragaria* genus (Fig. S10-14, Qiao et al. 2021).

Comment:

Demographic analyses assume absence of selection and population structure. However, as reported by the authors, this could not be the case specially (but not limited to) when the two main groups are analyzed ($k=2$). What impact could this have on the estimated demographic parameters and demographic histories? The authors should discuss this aspect in depth.

Response:

We agree with the reviewer that demographic analyses are sensitive to the effects of selection and population structure, which may alter the evolutionary rate in different parts of the genome.

1. Selection

As shown earlier, we included the aspect of selection in the bootstrap section for the peripheral pattern (lines 395-406), where we emphasized the long-range effects of linked selection resulting from autogamy or asexual reproduction. In the concluding remarks, we also considered variation in evolutionary rates across different genomic regions and suggested that the age of a haplotype could be “corrected,” for example, by using haplotype-specific nucleotide divergence (K) between species (lines 568-573):

“From an evolutionary perspective, the ability to trace ancestral haplotypes to distinct historical periods, particularly when future work accounts for variable evolutionary rates across genomic

regions (i.e. haplotype-specific nucleotide divergence, K , between species), has the potential to provide new insights into genome evolution and climatic adaptation in temperate flora.”

2. Population structure

MSMC-IM takes into account ancestral population structure (Title of article: “Tracking human population structure through time from whole genome sequences”) by detecting isolation events and admixture pulses between a pair of samples (Fig. 3). This approach adjusts the timing of coalescent events, providing estimates of population separation through time. Initially, MSMC/MSMC2 inferred only the divergence time of population pairs based on the ratio of cross-coalescence to within-coalescence rates (Schiffels and Durbin 2014). MSMC-IM extended this framework to reveal ancestral population structure through cumulative migration probabilities (Wang et al. 2020). When the cumulative migration probability (M) goes below 0.999, some ancestral haplotypes have not been exchanged between populations, indicating persistent population structure and incomplete genetic mixing. Notably, some east–west population pairs, particularly those between the Iberian and Lithuanian/Romanian regions, show consistent evidence of initial divergence ($M < 0.999$) already during the MIS 10 glaciation (Fig. S15A), with some pairs exhibiting even deeper structure ($M < 0.99$) at that time (Fig. S15B, Supplementary Data 4). However, even if population structure initially formed when populations sought western and eastern refugia during past glaciations, subsequent interglacial periods extensively mixed the eastern and western gene pools by secondary contacts (Fig. 3D–F, S12–S14), as also seen today through central Europe (Fig. 4B) and were reflected in the relatively high cumulative migration probabilities during earlier G-IG cycles (Fig. S15). Consequently, only relatively short fragments of the diverged haplotypes have accumulated in western (or peripheral) genomes from earlier G-IG cycles due to repeated episodes of secondary contacts. These ancient haplotypes may play an important role in climatic adaptation, potentially being resistant to genetic rescue (Fitzpatrick et al. 2020) and appear to have persisted through several G-IG cycles. From an evolutionary-genomic perspective, core populations with a large effective size are expected to dominate species-wide genome evolution by repeatedly reseeding peripheral populations that accumulate deleterious mutations (Peischl et al. 2013, Willi et al. 2018) either through genetic rescue (Fitzpatrick et al. 2020) or genetic swamping (Todesco et al. 2016, Kottler et al. 2021).

Other points

As suggested by the referee#2 the authors calculated a correlation between the proportions of admixture and some environmental variables. The environmental variables used have a very specific timeframe. Since the admixture events could be traced back a long time ago especially in a species with autogamous and asexual reproduction, I was asking what do these correlations refer to? i think the point should be clarified by the authors.

Response:

This was an important point that should be clarified in the Discussion. The availability of paleoclimate data is not sufficient to directly show the preservation of the European west-east temperature seasonality gradient over the entire timeframe of our study (400 ka). However, ample paleoclimate data, as well as ensembles of equilibrium climate simulations, are available for the time windows focused by the Paleoclimate Modelling Intercomparison Project (PMIP). These include the warmest stage of the current interglacial (6 ka), as well as last glacial maximum period (21 ka). We can draw on these results to estimate the likelihood of

the preservation of the seasonality gradient over glacial-interglacial cycles that showed the highest correlation with the eastern admixture proportion.

For 21 ka, a recent data-model synthesis (Cleator et al. 2020) shows a broadly uniform, deep negative anomaly in mean temperature of the coldest month in Europe. For mean temperature of the warmest month, the negative anomaly is smaller, but again broadly uniform over Europe, apart from northern European Russia where the negative anomaly is smaller. These patterns imply that at 21 ka, temperature seasonality in Europe was generally greater than today in all regions, but with no change in the west-east gradient of seasonality, except perhaps towards Russia where the seasonality gradient may have been even steeper than today.

For 6 ka, a paleodata synthesis and model simulations are shown separately in Mauri et al. (2014). The simulations show a mild positive anomaly in summer temperature spanning all of Europe, and virtually no change in winter temperature, implying no change in the east-west seasonality gradient. The paleodata for 6 ka show a distinctly different temperature pattern compared to the simulations, with warm anomalies in northern Europe and cold anomalies in southern Europe (in both seasons), which the authors attribute to a change in NAO/AO-type (North Atlantic Oscillation/Arctic Oscillation=NAO/AO) circulation which is not reproduced in the simulations. The winter temperature in the paleodata (but not simulations) also show an increase in anomalies towards eastern Europe, with northern European Russia and the Caucasus region warming by up to 5°C while the Ireland and western Iberia cool by 2°C. With no clear east-west gradient in the summer anomalies reconstructed from paleodata, this would imply a weakening of east-west seasonality gradient by about 7°C. This is, however, not nearly enough to abolish the eastward-increasing gradient of seasonality, as the difference in temperature seasonality is today around 20°C between the Atlantic seaboard of Europe and European Russia.

Overall, in the data and simulations covering the most recent transition from ice-age maximum conditions (21 ka) to peak interglacial conditions (6 ka), we find possible indications of both a steepening (21 ka data-model synthesis) and weakening (6 ka data) of the European west-east seasonality gradient compared to today. However, we find no indication of a weakening nearly strong enough to abolish the gradient, suggesting a persistence of the eastward-increasing seasonality gradient. When it comes to earlier G-IG cycles, given that they have been driven by the same forcings (Milankovitch insolation cycles) and affected by broadly consistent major feedbacks (ice sheet formation at similar locations and greenhouse gas variations between persistent boundary values), we would expect the 21 ka, 6 ka, and present-day snapshots to also broadly represent the situation in earlier interglacial and glacial stages.

We added the following clarification about persistent climatic differences between western and eastern European populations through time (lines 245- 248):

“Owing to their positions relative to the North Atlantic, the climatic differences likely persisted throughout GI cycles (Mauri et al. 2014, Cleator et al. 2020), suggesting that they contributed to genomic differentiation in woodland strawberry (Fig. 1D, 1F, S4, S5) and other species”.

Comment:

I think the authors should discuss in more detail what impact the possibility of autogamy and asexual reproduction might have had on the different analyses and what biases they might

have introduced into the results.

Response:

First, MSMC2 was developed for obligately outcrossing species. For that reason, we selected the samples with the lowest inbreeding coefficients from each region (Fig. S25-26) as our primary dataset for demographic modelling (Fig. 3, S19). These sample combinations consistently resulted in robust demographic trajectories matching with climatic events (Text S1). However, similar trajectories were visible also in many other sample combinations (Supplemental data 4).

Autogamy and asexual reproduction are expected to reduce the effective recombination rate relative to the mutation rate in populations, resulting in a lower ρ/θ ratio, where $\rho = 4N_e r$ and $\theta = 4N_e \mu$. When the recombination rate is not fixed in MSMC2, the program estimates both the effective recombination rate (ρ) and the mutation rate parameter (θ) directly from phased haplotypes, ρ after iterations. Importantly, a lower ρ/θ ratio (<1) is generally less problematic for demographic inference than a high ratio, because historical recombination events remain detectable due to the sufficient amount of mutations (Sellinger et al. 2021). In contrast, when the ρ/θ ratio is high (i.e. 10), recombination becomes difficult to detect as there are too few mutations marking recombination breakpoints, which can bias coalescent-based models and limit resolution in reconstructing the Ancestral Recombination Graph (ARG). We added the following paragraph to M&M to consider other modes of reproduction (lines 734-749).

“Since the mode of reproduction and N_E varies between populations, we did not use a fixed recombination rate in our MSMC2 analyses. Instead, population-scaled recombination rates (ρ) and mutation rates (θ) were inferred directly from the haplotype data, with the ratio of ρ to θ expected to be lower in populations with high levels of selfing or asexual reproduction. A low ρ/θ ratio is usually not problematic for coalescent-based models (except in the case of linked selection biasing demographic modelling), because it improves the detectability of historical recombination events within the mutational landscape, a key prerequisite for reconstructing the Ancestral Recombination Graph (ARG) (Sellinger et al. 2021). However, because MSMC2 was developed for obligately outcrossing species (Schiffels and Durbin 2014), an assumption not met in this species with a low heterozygosity due to ability to self, which may bias the inference (Sellinger et al. 2020), we selected from each region the samples with the lowest inbreeding coefficients (F_{ROH}) and/or the highest heterozygosity (to maximize the number of informative recombination events) as our primary dataset ($N = 41$; Fig. S25, S26) for demographic modelling.”

In non-optimal habitats, asexual reproduction may prevail and bias generation time estimates and may contribute to our observation that samples with the lowest inbreeding coefficients (Fig. S25) produced the most consistent and accurate temporal demographic trajectories (Fig. 3, S19). However, over evolutionary timescales, asexual reproduction likely has limited impact on the evolution of the perennial herb genome, as it leads to the accumulation of deleterious mutations through Muller’s ratchet (Muller 1964). For example, we observed that in peripheral regions in Alta and Kåfjord, the ratio of nonsynonymous to synonymous nucleotide diversity (π_a/π_s) is around 0.6, compared to approximately 0.4 in other regions. This suggests that purifying selection is less effective at removing deleterious nonsynonymous mutations in these peripheral edge populations as has been demonstrated in other perennial herb (Willi et al. 2018).

Therefore, we expect that source populations, particularly those with the highest effective population sizes, play a dominant role in shaping the long-term genomic evolution of the species. Only short genomic fragments have survived from peripheral populations from earlier G-IG-cycles, which is supported by the observation that much of the genomic content in peripheral populations consists of segments derived from source populations, based on relatively recent split times (Fig. S24) and high cumulative migration probabilities during earlier G-IG cycles (Fig. S15). Moreover, as mentioned earlier, effective population sizes have been relatively high ($N_E > 20000$) before the MIS2 suggesting that outcrossing has been prevalent mode of reproduction in this species, or at least the species ancestral haplotypes are descendants mostly from outcrossing populations, as would be expected, except during glaciations, when reproduction in peripheral populations may have occurred through autogamy or asexual means. Thus, parameter choices based on a perennial herb with sexual reproduction and a two-year generation time in optimal habitats are justified for demographic inference in this species as in other perennial herbs. While asexual reproduction may locally and transiently alter generation time in peripheral populations, such lineages are likely evolutionary dead ends (or form cryptic refugia) with limited influence on the species' long-term genomic trajectory, as selection operates inefficiently in the absence of recombination. The same largely applies to autogamous plant populations: like selfing species, they represent evolutionary tips and are often considered evolutionary dead ends. Taken together, sexual reproduction by outcrossing is expected to be the primary driving force of evolution in this perennial herb, as well as in many other species, even if currently most populations are much smaller than during earlier interglacial periods (Fig. S10).

Comment:

Apply corrections to significance levels when performing multiple (non-independent) tests

Response:

Bonferroni corrections have been applied to admixture proportion - environmental variable correlations (Fig. 1F, S4). FDR values were applied for p-values using the Benjamini-Hochberg method regarding the significance of split times (Supplementary Data 7, Fig. S24) and IBD correlations (Fig. 1E).

Final recommendation

Overall, I believe that the article deserves to be accepted, but only after the authors clarify the above points and correct several minor details, errors and repetitions in the text. For example, the name of the author of a species should be provided at least the first time the species is named....

Authors are encouraged to reread the entire article carefully before resubmitting it.

Response:

We have reread the article carefully, removed repetitions, provided the authors of species, and improved the language throughout the article.

References:

Charlesworth B, Morgan MT, Charlesworth D. 1993. The effect of deleterious mutations on neutral molecular variation. *Genetics* 134:1289-303.

Cleator SF, Harrison SP, Nichols NK, Prentice IC, Roulstone I. 2020. A new multivariable benchmark for Last Glacial Maximum climate simulations. *Clim. Past* 16:699-712.

Drovetski SV, Fadeev IV, Raković M, Lopes RJ, Boano G, et al. 2018. A test of the European Pleistocene refugial paradigm, using a Western Palaearctic endemic bird species. *Proceedings of the Royal Society B: Biological Sciences* 285:20181606.

Fitzpatrick SW, Bradburd GS, Kremer CT, Salerno PE, Angeloni LM, Funk WC. 2020. Genomic and Fitness Consequences of Genetic Rescue in Wild Populations. *Current Biology* 30:517-22.e5.

Gillespie JH. 2000. Genetic Drift in an Infinite Population: The Pseudohitchhiking Model. *Genetics* 155:909-19.

Hewitt GM. 1999. Post-glacial re-colonization of European biota. *Biological Journal of the Linnean Society* 68:87-112.

Hewitt GM. 2000. The genetic legacy of the Quaternary ice ages. *Nature* 405:907-13.

Hewitt GM. 2011. Mediterranean peninsulas: the evolution of hotspots. In: Zachos FE, Habel JC (eds) Biodiversity hotspots. Springer, Berlin, pp 123–147.

Hill WG, Robertson A. 1966. The effect of linkage on limits to artificial selection. *Genetical Research*, 8:269–294.

Johri P, Riall K, Becher H, Excoffier L, Charlesworth B, Jensen JD. 2021. The Impact of Purifying and Background Selection on the Inference of Population History: Problems and Prospects. *Molecular Biology and Evolution* 38:2986-3003.

Keightley PD, Otto SP. 2006. Interference among deleterious mutations favours sex and recombination in finite populations. *Nature* 443:89-92.

Kottler EJ, Dickman EE, Sexton JP, Emery NC, Franks SJ. 2021. Draining the Swamping Hypothesis: Little Evidence that Gene Flow Reduces Fitness at Range Edges. *Trends in Ecology & Evolution* 36:533-44.

Laenen B, Tedder A, Nowak MD, Toräng P, Wunder J, Wötzel S, Steige KA, Kourmpetis Y, Odong T, Drouzas AD, et al. 2018. Demography and mating system shape the genome-wide impact of purifying selection in *Arabidopsis thaliana*. *Proceedings of the National Academy of Sciences of the United States of America* 115:816-821.

Mauri A, Davis BAS, Collins PM, Kaplan JO. 2014. The influence of atmospheric circulation on the mid-Holocene climate of Europe: a data–model comparison. *Clim. Past* 10:1925-38.

Mergeay J, Smet S, Collet S, Nowak S, Reinhardt I, Kluth G, Szewczyk M, Godinho R, Nowak C, Mysłajek RW, Rolshausen G. 2024. Estimating the Effective Size of European Wolf Populations. *Evolutionary Applications*. Oct 22;17(10): e70021.

Mora-Carrera E, Stubbs RL, Potente G, Yousefi N, Aeschbacher S, Keller B, Choudhury RR, Celep F, Kochjarová J, de Vos JM, Szövényi P, Conti E. 2024. Unveiling the Genome-Wide Consequences of Range Expansion and Mating System Transitions in *Primula vulgaris*. *Genome Biology and Evolution* 16(10), evae208.

Muller HJ. 1964. The relation of recombination to mutational advance. *Mutation Research* 106:2-9.

Peischl S, Dupanloup I, Kirkpatrick M, Excoffier L. 2013. On the accumulation of deleterious mutations during range expansions. *Molecular Ecology* 22:5972-82.

Schrider DR, Shanku AG, Kern AD. 2016. Effects of Linked Selective Sweeps on Demographic Inference and Model Selection. *Genetics* 204:1207-23.

Sellinger TPP, Abu Awad D, Moest M, Tellier A. 2020. Inference of past demography, dormancy and self-fertilization rates from whole genome sequence data. *PLoS Genet* 16(4): e1008698.

Sellinger TPP, Abu-Awad D, Tellier A. 2021. Limits and convergence properties of the sequentially Markovian coalescent. *Molecular Ecology Resources* 21:2231-48.

Smith JM, Haigh J. 1974. The hitch-hiking effect of a favourable gene. *Genetical Research* 23:23-35.

Todesco M, Pascual MA, Owens GL, Ostevik KL, Moyers BT, Hübner S, Heredia SM, Hahn MA, Caseys C, Bock DG, Rieseberg LH. 2016. Hybridization and extinction. *Evolutionary Applications*. 22;9(7):892-908.

Wang K, Mathieson I, O'Connell J, Schiffels S. 2020. Tracking human population structure through time from whole genome sequences. *PLOS Genetics* 16 (3): e1008552.

Willi Y, Fracassetti M, Zoller S, Van Buskirk J. 2018. Accumulation of Mutational Load at the Edges of a Species Range. *Molecular Biology and Evolution* 35:781-91.

Willis KJ, Bennett KD, Walker D, Hewitt GM. 2004. Genetic consequences of climatic oscillations in the Quaternary. *Philosophical Transactions of the Royal Society of London. Series B: Biological Sciences* 359:183-95.